# RWKVQuant: Quantizing the RWKV Family with Proxy Guided Hybrid of Scalar and Vector Quantization

**Chen Xu** [* 1]  **Yuxuan Yue** [* 1 2]  **Zukang Xu** [1]  **Xing Hu** [1]
**Jiangyong Yu** [1]  **Zhixuan Chen** [1]  **Sifan Zhou** [1]  **Zhihang Yuan** [1]  **Dawei Yang**[✉ 1]

## Abstract

RWKV is a modern RNN architecture with comparable performance to Transformer, but still faces challenges when deployed to resource-constrained devices. Post Training Quantization (PTQ), which is a an essential technique to reduce model size and inference latency, has been widely used in Transformer models. However, it suffers significant degradation of performance when applied to RWKV. This paper investigates and identifies two key constraints inherent in the properties of RWKV: (1) Non-linear operators hinder the parameter-fusion of both smooth- and rotation-based quantization, introducing extra computation overhead. (2) The larger amount of uniformly distributed weights poses challenges for cluster-based quantization, leading to reduced accuracy. To this end, we propose RWKVQuant, a PTQ framework tailored for RWKV models, consisting of two novel techniques: (1) a coarse-to-fine proxy capable of adaptively selecting different quantization approaches by assessing the uniformity and identifying outliers in the weights, and (2) a codebook optimization algorithm that enhances the performance of cluster-based quantization methods for element-wise multiplication in RWKV. Experiments show that RWKVQuant can quantize RWKV-6-14B into about 3-bit with less than 1% accuracy loss and 2.14× speed up.

## 1. Introduction

RWKV (Peng et al., 2023) is a modern sequence model that integrates the strengths of both Recurrent Neural Networks (RNNs) (Elman, 1990) and Transformer (Vaswani, 2017).

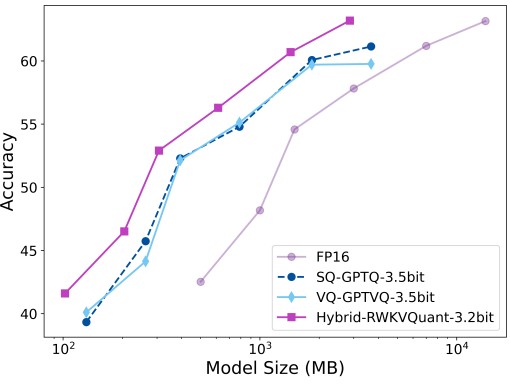

Figure 1: Accuracy-model size curve. Results of zero-shot accuracy are evaluated on the LAMBADA dataset (Radford et al., 2019). Our proposed RWKVQuant outperforms the individual utilization of SQ and VQ methods for all sizes of models.

It has a comparable capacity to Transformer-based Large Language Models (T-LLMs) while retaining the efficient inference feature of RNNs, positioning it a promising foundational architecture for both language (Peng et al., 2024) and vision (Zhou & Chen, 2024) tasks.

Despite the advantages, its vast size of parameters have posed a significant barrier to the deployment on resource-constrained devices. For instance, the compute-to-memory access ratio (FLOPs/Bytes) of RWKV-6-7B (Peng et al., 2024) is 0.97, while that for the decoding phase of LLaMA-2-7B (Touvron et al., 2023) is 4.88 (detailed in A.3). Secondly, large RWKV models demand substantial memory resources. For instance, RWKV-6-14B requires approximately 30GB of memory to be loaded, which typically exceeds the capacity of edge devices.

Post Training Quantization (PTQ), including Scaler Quanzization (SQ) and Vector Quantization (VQ), is a widely adopted approach to reduce model size and inference latency for T-LLMs (Shao et al., 2023; Ashkboos et al., 2024; ost, 2024; Yuan et al., 2024). However, directly applying the most advanced quantization frameworks to RWKV models leads to severe performance degradation. For instance,

---

[*]Equal contribution  [1]Houmo AI  [2]Harbin Institute of Technology (Shenzhen).  Correspondence to: Dawei Yang <dawei.yang@houmo.ai>.

*Proceedings of the 42nd International Conference on Machine Learning*, Vancouver, Canada. PMLR 267, 2025. Copyright 2025 by the author(s).

Table 1: The average relative cluster loss of weights for the RWKV and LLaMA family, computed by KMeans (Lloyd, 1982).

| Family | Model | 8 Clusters | 16 Clusters |
|---|---|---|---|
| RWKV | 6-7B | 2.01 | 0.78 |
| | 6-14B | 1.98 | 0.78 |
| LLaMA | 2-7B | 0.96 | 0.65 |
| | 2-14B | 0.89 | 0.64 |

applying QuaRot (Ashkboos et al., 2024) (belongs to SQ) to RWKV-7 series models increases the overall FLOP by more than 99%, and applying VPTQ (Liu et al., 2024a) (belongs to VQ) to RWKV-6-7B model leads to more than 16% accuracy decline.

In depth, we investigate and identify two primary limitations inherent in the properties of RWKV. ❶ **Non-linear operators hinder the parameter-fusion of both smooth- and rotation-based methods.** Typically, these SQ approaches fuse the introduced parameters, i.e., smoothing vectors and orthogonal matrices, into neighbored normalization layers and linear layers of T-LLMs. However, the RWKV structure employs several non-linear operators along the fusion path, including token-shift, Sigmoid function, and exponential function. These modules can block the linear fusion process, inevitably leading to additional runtime overhead. ❷ **The larger amount of uniformly distributed weights poses challenges for cluster-based quantization.** While such VQ methods benefit from distinctly categorized distribution, RWKV tends to have more uniform weights compared to T-LLMs (detailed in Section 4.4), which complicates the clustering process as shown in Table 1.

To this end, we propose RWKVQuant, an effective and efficient post-training quantization (PTQ) framework tailored for RWKV models. Our core insight is to enhance VQ by partially applying the classic compensation-based SQ methods like GPTQ (Frantar et al., 2022), which are more suitable for uniformly distributed weights. Specifically, we propose a coarse-to-fine proxy to optimize the hybrid strategy. (1) The coarse-grained proxy is established on the basis of Information Entropy (IE) (Shannon, 1948), which evaluate the overall uniformity. For non-uniform weights, VQ is directly applied. (2) For uniform weights, we further introduce a fine-grained proxy, computed by weighted high-order central moments, to detect local outliers. VQ is applied when outliers emerge; otherwise, SQ is applied. In addition to the hybrid, we further optimize VQ for the unique element-wise multiplication operator of RWKV.

Experiments show that RWKVQuant outperforms existing methods across various tasks on different RWKV model

families, including RWKV-6 (Peng et al., 2024) and RWKV-7 (Bo, 2021) for RWKV-based language tasks, as well as VRWKV (Duan et al., 2024) for RWKV-based vision tasks. As shown in Figure 1, RWKVQuant quantizes weights into about 3-bit and achieves superior accuracy compared to the individual utilization of SQ and VQ. Additionally, RWKVQuant demonstrates remarkable efficiency. For instance, it can quantize RWKV-6-14B with less than 1% accuracy loss, $2.83\times$ memory saving, and $2.14\times$ speed up. Lastly, our contributions can be concluded as follows.

- We reveal that both smooth- and rotation-based PTQ methods are not well-suitable for RWKV, primarily due to the unavoidable runtime overhead. Further, cluster-based PTQ methods suffer severe accuracy drop, owing to the larger amount of uniformly distributed weights.

- We propose RWKVQuant, which enhances VQ by partially adopting compensation-based SQ methods. It introduces a coarse-to-fine proxy to guide the hybrid strategy. It further enhances the VQ for the unique element-wise multiplication modules in RWKV.

- RWKVQuant can effectively and efficiently quantize weights into about 3-bit and outperforms both SQ and VQ methods as shown in Figure 1.

- To the best of our knowledge, RWKVQuant is the first comprehensive PTQ framework for the RWKV family. As a pioneering study, we will publish the code in the hope of promoting further research and facilitating advancements in this field.

## 2. Preliminaries

### 2.1. RWKV Structure and Models

Referring to Figure 2, RWKV structure contains two key modules, including Time Mixing and Channel Mixing (detailed in A.1). With the previous word $x_{t-1}$, the current word $x_t$ can be derived by a token-shift operator:

$$x_t = \text{concat}(x_{t-1}[1:,], \mathbf{0}), \quad (1)$$

where $\mathbf{0}$ denotes an all-zero vector. RWKV models make use of Time Mixing for seizing the relationship among tokens and utilize Channel Mixing to probe the dimensions within the hidden layer that are relevant to individual tokens.

Compared to T-LLMs, these modifications enables RWKV to decrease substantially the computational overhead and memory demands while effectively retaining the capacity to model long-term dependencies. Thereby, the RWKV family has already manifested its potential in a diverse array of real-world applications (Li et al., 2024), including QQ (Cryscan, 2023), WeChat (MrTom34, 2023; LeoLin4258, 2024), and Telegram (spion, 2023). For natural

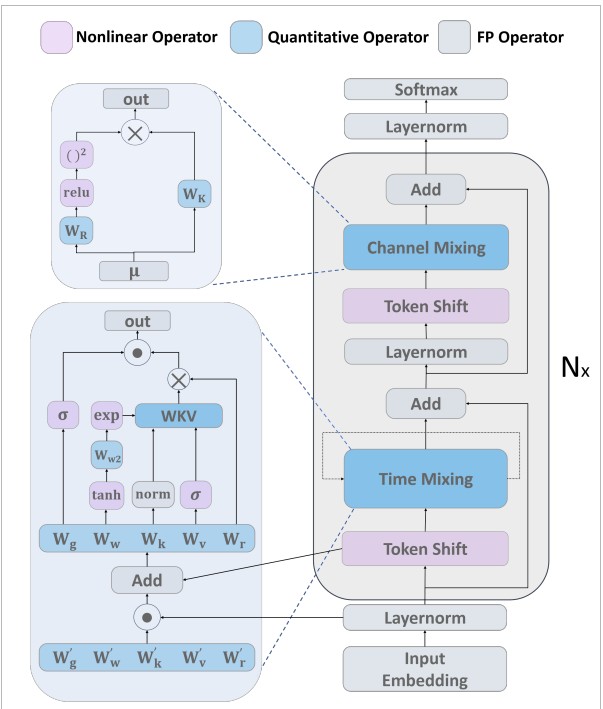

Figure 2: Model Structure of RWKV-7. It contains several blocks and each block has two key modules: Time Mixing and Channel Mixing.

language understanding tasks, RWKV-v6 (Peng et al., 2024) has achieved remarkable advancements in accuracy, attaining comparable performance to those of larger models like SomlLM and Qwen (Chen et al., 2024). For vision tasks, BSBP-RWKV (Zhou & Chen, 2024) excels in the domain of image segmentation.

## 2.2. Post Training Quantization (PTQ)

PTQ serves as a potent strategy for model compression. By converting the high-precision variables of pre-trained models into low-bit integers, it achieves a reduction in memory usage and an acceleration of inference speed. Typically, PTQ can be divided into two main approaches: SQ and VQ.

**Scalar Quantization (SQ)**  SQ maps the original data to the quantized range by a scaling factor, and subsequently rounds floating points to the uniform-distributed integers. For a tensor $x$ to be quantized, it can be uniformly quantized to $b$-bits as follows (Jacob et al., 2018):

$$Q(\boldsymbol{x}) = \text{clamp}(\lfloor \frac{\boldsymbol{x}}{s} \rceil + z, 0, 2^b - 1), \quad (2)$$

where $Q(\cdot)$ represents the quantization function, $s = (\max(\boldsymbol{x}) - \min(\boldsymbol{x}))/2^b - 1$ is the scale factor, $z = -\min(\boldsymbol{x})/s$ is the zero point, $\lfloor \cdot \rceil$ denotes the rounding-to-nearest operator, and clamp is the clipping function.

SQ is widely-adopted by most of the PTQ frameworks (Yang et al., 2024). For instance, the classic compensation-based GPTQ (Frantar et al., 2022) can quantize weights to 3-4 bit with slight accuracy drop based on approximate second-order information. To address outliers, AWQ (Lin et al., 2023), SmoothQuant (Xiao et al., 2022), and Omni-Quant (Shao et al., 2023) explore the scheme of smoothing by detecting the importance of different activation channels. Recent works (e.g., Quarot (Ashkboos et al., 2024), Spin-Quant (Liu et al., 2024b), and OSTQuant (ost, 2024)) further suppress outliers by rotating the variables to be quantized with orthogonal matrices.

**Vector Quantization (VQ)**  VQ quantizes several vectors into a finite subset, which is commonly referred to as a codebook $\boldsymbol{C}$ (Gersho, 1979). Typically it has shape $(2^k, d)$, where $k$ is the bits of the index and $d$ is the vector dimension. Given a tensor $\boldsymbol{x}$ with shape $(m, n)$ to be quantized, VQ first transforms it into $x'$ with dimensions $(m * n//d, d)$. Second, for each $d$-dimensional vector in $\boldsymbol{x}'$, VQ replaces it with the $k$-bit index of the nearest vector from the codebook. For instance, if we use the Euclidean distance (calculated by the Frobenius normalization $||\cdot||_F$) to measure similarities, the quantization process can be expressed as:

$$Q(\boldsymbol{x}') = \{\underset{j \in 2^k}{\arg\min}||\boldsymbol{x}'_i - \boldsymbol{C}_j||_F \,|\, i = 1, ..., m*n//d\}. \quad (3)$$

Compared to SQ, this scheme takes the advantage of maintaining the shape of the source distribution, especially under lower bit-width. For example, VPTQ (Liu et al., 2024a) and GPTVQ (van Baalen et al., 2024) combine VQ with GPTQ, achieving advanced performances under 2~3 bits. To obtain the codebook, they cluster the source vectors by K-Means Algorithm (Lloyd, 1982) and Expectation-Maximization Algorithm (Moon, 1996), respectively. AQLM (Egiazarian et al., 2024) further utilizes layer-wise training for the codebook to obtain optimal accuracy.

## 3. Method

### 3.1. Coarse-to-fine Proxy for Hybrid Quantization

**Hybrid of SQ and VQ**  Given inputs $\boldsymbol{x}$, weights $\boldsymbol{\theta}$, number of weights $M$, and the model $f(\cdot)$, the optimization goal is to minimize the expectation $\mathbb{E}[\cdot]$ of the Mean Square Error (MSE) of the model output:

$$\begin{aligned} &\underset{\boldsymbol{\phi}}{\arg\min} \, \mathbb{E}[\,||f_{\boldsymbol{\theta}}(\boldsymbol{x}) - f_{\boldsymbol{\theta}'}(\boldsymbol{x})||_F^2\,] \\ &\text{s.t.} \, \boldsymbol{\phi} = \{\phi_m \in \{0, 1\} \,|\, m = 1, 2, ..., M\} \\ &\quad \boldsymbol{\theta}' = \{\phi_m \text{SQ}(\boldsymbol{\theta}_m) + (1 - \phi_m)\text{VQ}(\boldsymbol{\theta}_m) \\ &\quad\quad |\, \boldsymbol{\theta}_m \in \boldsymbol{\theta}, m = 1, 2, ..., M\}. \end{aligned} \quad (4)$$

Here, $\boldsymbol{\phi}$ represents the collection of options for SQ and VQ. Although the optimal solution of Equation 4 can be found by

the exhaustive algorithm, its complexity increases exponentially with the number of weights, i.e., $O(2^M)$. Considering the computational cost, we construct an effective proxy by evaluating the uniformity and outliers of each weight from both coarse- and fine-grained perspectives, whose complexity decreases to $O(M)$.

**Coarse-grained Proxy** Information Entropy (IE) (Shannon, 1948) is one of the most common approaches to evaluate uniformity. However, it measures the probability distribution, rather than the original data that concerned by quantization. To take advantage of its effectiveness, we perform a series of transformations on the model weights.

Given a weight $W \in \mathbb{R}^{oc \times ic}$, it is first flattened, then sorted in ascending order to formulate $W' \in \mathbb{R}^{oc \cdot ic}$. Subsequently, the intervals $G \in \mathbb{R}^{(oc \cdot ic)-1}$ of all adjacent positions in $W'$ can be calculated by:

$$G = W'[1:] - W'[:-1]. \tag{5}$$

For the clarity of expression, the term $(oc \cdot ic) - 1$ is denoted by the symbol $n$ in the following contents. Next, $G$ is transformed to $G'$ by:

$$G' = \{G'_i = \frac{G_i}{\sum_{i=1}^{n} G_i} \mid i = 1, 2, ..., n\}. \tag{6}$$

Considering that $G'$ satisfies that $\sum_{i=1}^{n} G'_i = 1$, it can be treated as a discrete probability distribution. Consequently, its IE (denoted by $H$) can be obtained by:

$$H(G') = -\sum_{i=1}^{n} G'_i \log G'_i. \tag{7}$$

According to the property of IE, Equation 7 measures the concentration of $G'$. Since $G'$ are the intervals, its concentration can equivalently reflect the uniformity of the original weight $W$.

Assuming an absolutely uniform weight $\hat{W}$ with fixed intervals, it can be transformed to $\hat{G}'$ following the above process, which finally should be:

$$\hat{G}' = \{\hat{G}'_i = \frac{1}{n} \mid i = 1, 2, ..., n\}. \tag{8}$$

Owing to the property of IE, only if $W = \hat{W}$ does Equation 7 take the maximum value. Finally, the coarse-grained proxy $P_c$ can be obtained by computing the gap between the IE of $G'$ and $\hat{G}'$:

$$P_c(G') = H(\hat{G}') - H(G'). \tag{9}$$

By introducing a threshold $\tau_c$, non-uniform weights can have larger values of $P_c$, indicating the usage of VQ as shown in Figure 3(a). Since IE is a measure of the entire

system, a small amount of local outliers does not significantly effect $P_c$. However, in case of SQ, the accuracy is highly dependent to the data scale. Such outliers can cause more minimal values to be mapped to the same integer, thus increasing the rounding error. For instance, Figure 3(b) and Figure 3(c) have close $P_c$ values, while the former contains obvious outliers and is more accurate under VQ.

**Fine-grained Proxy** To mitigate the issue that $P_c$ is not sensitive enough to local outliers of a relatively uniform data, we further introduce a fine-grained proxy. Specifically, we perform the Taylor expansion (Taylor, 1717) to Equation 9 to evaluate the minor disturbances $\delta$ around $\hat{G}'$.

*Step 1* The gap $\delta$ between $G'$ and $\hat{G}'$ can be written as:

$$\delta = G' - \hat{G}' = \{\delta_i = G'_i - \frac{1}{n} \mid i = 1, 2, ..., n\}. \tag{10}$$

According to Equation 6, it should be satisfied that:

$$\sum_{i=1}^{n} \delta_i = \sum_{i=1}^{n} (G'_i - \frac{1}{n}) = 0. \tag{11}$$

*Step 2* The Taylor expansion can be formulated as:

$$P_c(G') = P_c(\hat{G}') + \sum_{k=1}^{K} (k!)^{-1} P_c^k(\hat{G}') \delta^k + o(\delta^K)$$

$$= \sum_{k=1}^{K} (k!)^{-1} \sum_{i=1}^{n} \frac{\partial^k P_c}{\partial G'^k_i} \bigg|_{G'_i = \frac{1}{n}} \delta_i^k + o(\delta^K). \tag{12}$$

Taking the Euler's number $e$ as the base of the $\log$ function in Equation 9, the $k$-th order partial derivative of $P_c$ with respect to $G'_i$ can be expressed as:

$$\frac{\partial^k P_c}{\partial G'^k_i} = \begin{cases} \ln G'_i + 1 & k = 1 \\ (-1)^k (k-2)! G'^{(1-k)}_i & k \geq 2 \end{cases}. \tag{13}$$

*Step 3* Taking Equation 11 and 13 into consideration, Equation 12 can be transformed into:

$$P_c(G') = \sum_{k=2}^{K} \frac{(-1)^k n^{k-1}}{k(k-1)} \sum_{i=1}^{n} \delta_i^k + o(\delta^K). \tag{14}$$

*Step 4* Omitting the term $o(\delta^K)$, Equation 14 can be reformulated as:

$$P_c(G') \approx [s_2, ..., s_K] \odot [v_2, ..., v_K] \odot [M_2, ..., M_K],$$

$$\text{where} \quad s_k = (-1)^k, \ v_k = \frac{n^k}{k(k-1)}, \ M_k = \frac{\sum_{i=1}^{n} \delta_i^k}{n}. \tag{15}$$

Here, '$\odot$'represents element-wise multiplication. $M_k$ is the $k$-th order central moment of $G'$, which is defined as:

$$M_k(G') = \mathbb{E}[(G' - \mathbb{E}[G'])^k]. \tag{16}$$

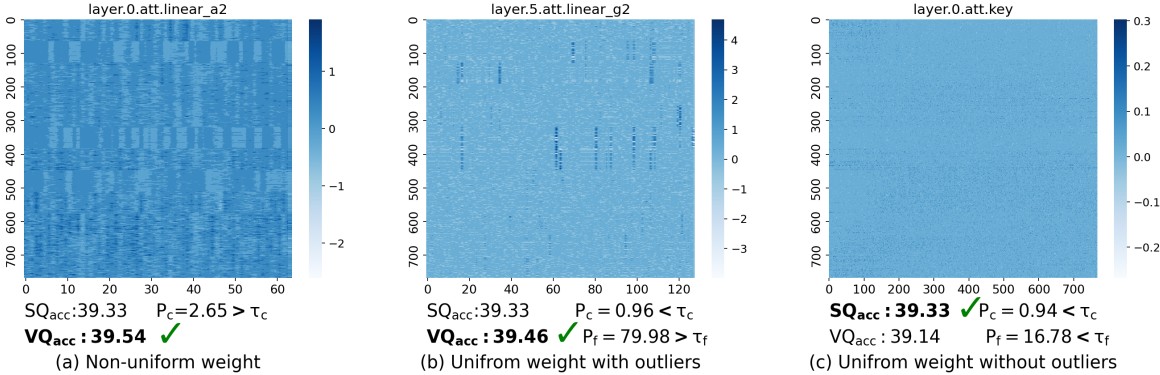

Figure 3: Zero-shot accuracy when applying different quantization methods to specific weights. For the weight in each sub-figure, $SQ_{acc}$ denotes the performance when SQ is applied, $VQ_{acc}$ denotes the performance when VQ is applied, while all other weights are quantized with VQ. $P_c$ and $P_f$ are coarse- and fine-grained proxy, while $\mu_c$ and $\mu_f$ are their corresponding thresholds.

The central moment can serve as a metric for assessing the local features of data. This is because the difference between outliers and other data points is magnified by the $k$-th power. For instance, when $k = 2$, Equation 16 yields the variance, indicating the spread of the data. For $k = 3$, the central moment corresponds to skewness, assessing the symmetry of the data. For $k = 4$, it represents kurtosis, revealing the data's long-tail characteristic.

_Step 5_ Considering that $P_c(\boldsymbol{G}')$ signifies the overall uniformity, $s_k$ and $v_k$ can be regarded as the direction and the significance of the local feature $M_k$. Since only the magnitudes of features are considered when it comes to outliers, the fine-grained proxy can be defined as:

$$P_f(\boldsymbol{G}') = \sum_{k=2}^{K} v_k |M_k|, \tag{17}$$

where $K$ is a hyper-parameter. By introducing a threshold $\tau_f$, outliers can be detected by larger values of $P_f$, indicating the usage of VQ as shown in Figure 3(b).

Finally, our proxy solution of Equation 4 can be obtained by the combination of $P_c$ and $P_f$:

$$\hat{\boldsymbol{\phi}} = \{\phi_z = \begin{cases} 1 & P_c(\boldsymbol{G}'_z) < \tau_c \text{ and } P_f(\boldsymbol{G}'_z) < \tau_f \\ 0 & P_c(\boldsymbol{G}'_z) < \tau_c \text{ and } P_f(\boldsymbol{G}'_z) \geq \tau_f \\ & \text{or } P_c(\boldsymbol{G}'_z) \geq \tau_c \end{cases}$$
$$\mid z = 1, 2, ..., M\}, \tag{18}$$

where $\boldsymbol{G}'_z$ denotes the $m$-th weight after the transformation. Only if both the coarse-grained proxy and the fine-grained proxy are lower than their corresponding threshold will SQ be applied, as shown in Figure 3(c). Otherwise, the weight distribution is supposed to be generally uneven, or relatively uniform but with local outliers, which indicates

the application of VQ. Notably, the fine-grained proxy is only utilized in condition that $P_c(\boldsymbol{G}'_z) < \tau_c$.

### 3.2. Codebook Optimization for Element-wise Multiplication

Different from Transformer-based LLMs, the element-wise multiplication '$\odot$'between the input $\boldsymbol{x}$ and the weight $\boldsymbol{\mu}$ is applied in all projection layers of the RWKV structure, as shown in Figure 2. In accordance with the proxy introduced in Section 3.1, VQ is expected to be applied to most of them. However, existing VQ methods are primarily tailored for matrix multiplication modules. We thereby propose to optimize the VQ codebook specifically for element-wise multiplication modules.

Given a weight $\boldsymbol{\mu} \in \mathbb{R}^{m \times n}$, it is first transformed to $\boldsymbol{\mu}' \in \mathbb{R}^{(m \cdot n // d) \times d}$, where $d$ is the hidden dimension. Following the VQ process stated in Equation 3, it can be quantized into $Q(\boldsymbol{\mu}')$. Typically, the quantization loss $\mathcal{L}$ can be written as:

$$\mathcal{L} = ||\boldsymbol{X} \odot \boldsymbol{\mu}' - \boldsymbol{X} \odot \text{Deq}(Q(\boldsymbol{\mu}'))||_F^2$$
$$= \sum_{i=1}^{m \cdot n // d} \sum_{j=1}^{d} \boldsymbol{X}_{ij}^2 (\Delta \boldsymbol{\mu}'_{ij})^2, \tag{19}$$

where $\boldsymbol{X}$ is a representative of the calibration activations, Deq is the de-quantization process, and $\Delta \boldsymbol{\mu}'$ denotes the quantization error of the weight. To minimize Equation 19, a larger activation value should correspond to a smaller $\Delta \boldsymbol{\mu}'$, indicating the significance of this position. Thus, we employ the term $\boldsymbol{X}^2$ to direct the weighted KMeans algorithm in the generalization of codebooks.

Due to the nature of element-wise multiplication, $\boldsymbol{X}$ must have the same shape as $\boldsymbol{\mu}'$, which further introduces an issue of integrating batches of data. The most straightforward approach is to simply average all samples. However,

Table 2: Comparison of perplexity on LAMBADA and averaged accuracy on nine Zero-Shot tasks. For all methods except ours and floating-point, we report metrics under both bpw settings of 3.5 and 3.25.

| Bpw. | Method | RWKV7-0.1B | | RWKV7-0.5B | | RWKV7-1.47B | | RWKV6-1B | | RWKV6-3B | | RWKV6-7B | | RWKV6-14B | |
|---|---|---|---|---|---|---|---|---|---|---|---|---|---|---|---|
| | | 0-shot[9] Avg.(↑) | LambA. (↓) | 0-shot[9] Avg.(↑) | LambA. (↓) | 0-shot[9] Avg.(↑) | LambA. (↓) | 0-shot[9] Avg.(↑) | LambA. (↓) | 0-shot[9] Avg.(↑) | LambA. (↓) | 0-shot[9] Avg.(↑) | LambA. (↓) | 0-shot[9] Avg.(↑) | LambA. (↓) |
| 16 | FloatingPoint | 43.02 | 14.21 | 48.67 | 7.21 | 55.08 | 4.80 | 54.39 | 4.60 | 58.32 | 3.83 | 61.69 | 3.21 | 63.65 | 3.02 |
| 3.25 | RTN | 36.22 | 152.82 | 39.99 | 57.11 | 45.46 | 11.43 | 49.84 | 6.39 | 54.17 | 4.71 | 58.34 | 3.87 | 61.16 | 3.34 |
| | GPTQ | 37.92 | 63.54 | 41.16 | 23.29 | 51.15 | 7.93 | 50.55 | 6.43 | 53.94 | 4.88 | 59.28 | 3.72 | 60.18 | 3.43 |
| | AWQ | 36.20 | 132.06 | 68.92 | 5.92 | 43.62 | 15.27 | 44.41 | 17.97 | 47.24 | 11.97 | 49.48 | 8.33 | 49.35 | 8.18 |
| | QuaRot | 34.53 | 243.99 | 40.17 | 76.89 | 50.81 | 9.39 | 45.02 | 29.38 | 48.24 | 22.67 | 54.29 | 8.81 | 54.76 | 14.05 |
| | kMeans | 38.21 | 87.06 | 44.59 | 20.19 | 52.77 | 6.57 | 47.02 | 15.93 | 53.06 | 8.27 | 55.72 | 4.69 | 61.40 | 4.61 |
| | GPTVQ | 40.25 | 23.75 | 43.64 | 14.15 | 52.06 | 5.54 | 49.86 | 6.11 | 54.23 | 4.31 | 58.57 | 3.49 | 59.63 | 3.15 |
| | VPTQ | 35.78 | 128.59 | 40.14 | 30.63 | 45.83 | 11.13 | 43.89 | 14.67 | 48.22 | 7.77 | 53.06 | 4.75 | 57.87 | 3.62 |
| 3.5 | RTN | 38.08 | 81.14 | 43.02 | 25.09 | 51.74 | 7.89 | **51.76** | 5.83 | 54.42 | 4.50 | 59.20 | 3.59 | 60.84 | 3.31 |
| | GPTQ | 39.33 | 40.16 | 45.73 | 13.07 | 52.28 | 6.55 | 51.23 | 5.86 | 54.89 | 4.54 | 60.07 | 3.68 | 61.14 | 3.29 |
| | AWQ | 38.31 | 55.72 | 42.40 | 16.98 | 44.61 | 10.71 | 51.20 | 10.07 | 48.40 | 8.77 | 50.50 | 7.07 | 50.86 | 6.06 |
| | QuaRot | 37.26 | 126.19 | 40.84 | 40.38 | 51.98 | 7.94 | 47.00 | 16.29 | 51.01 | 16.99 | 56.95 | 6.44 | 56.33 | 10.63 |
| | kMeans | 39.55 | 36.26 | 43.07 | 17.05 | **52.77** | 6.57 | 50.03 | 8.24 | 54.99 | 6.28 | 59.51 | 3.96 | 61.15 | 3.84 |
| | GPTVQ | 40.10 | 25.82 | 44.13 | 10.88 | 52.13 | 5.51 | 50.29 | 5.74 | 55.12 | 4.12 | 59.70 | 3.30 | 59.76 | 3.34 |
| | VPTQ | 37.07 | 74.70 | 41.06 | 25.03 | 47.38 | 9.52 | 43.82 | 14.74 | 48.86 | 8.62 | 52.95 | 4.47 | 57.93 | 3.75 |
| 3.275 | **Ours** | **41.10** | **18.41** | **46.01** | **9.39** | 52.40 | **5.24** | 51.69 | **5.29** | **55.79** | 3.88 | **60.19** | 3.23 | **62.69** | **2.89** |

this method is not sufficiently effective because it is highly susceptible to the influence of a small number of outliers. Given that the activations of RWKV models typically follows an approximately normal distribution, we introduce a percentile-based clipping operation to limit the range of samples prior to averaging, thereby alleviating this issue.

## 4. Experiments

### 4.1. Experimental Settings

**Models and Datasets.** We evaluate the RWKVQuant framework on RWKV6 (Peng et al., 2024), RWKV7, and VR-WKV models (Duan et al., 2024). For vision tasks, we utilize ImageNet (Deng et al., 2009) for image classification, Coco (Lin et al., 2014) for object detection, and ADE20K (Zhou et al., 2019) for segmentation. Aligned with the accuracy evaluation methods used in the VRWKV experiments, we report Top-1 Accuracy for classification tasks, Box Average Precision (AP) for detection tasks, and Mean Intersection over Union (MIoU) for segmentation tasks. For language tasks, consistent with the RWKV6 paper, we report the perplexity (PPL) on the Lambada dataset. We also evaluate the models on up to nine zero-shot tasks using the LM-evaluation-harness (version 0.4.4), including LAMBADA(OpenAI) (Radford et al., 2019), HEADQA (EN) (Rogers et al., 2023), HellaSwag (Zellers et al., 2019), OpenBookQA (OBQA) (Mihaylov et al., 2018), PIQA (Bisk et al., 2020), SCIQ (Pedersen et al., 2020), Winogrande (Sakaguchi et al., 2021), ARC-Challenge and ARC-Easy (Boratko et al., 2018).

**Baselines and Implementation Details.** In addition to comparing with SQ methods such as RTN, GPTQ (Frantar et al., 2022), AWQ (Lin et al., 2023), and Qurot (Ashkboos et al., 2024), we also benchmark our approach against VQ methods like K-Means, GPTVQ (van Baalen et al., 2024) and VPTQ (Liu et al., 2024a) for weight-only quantization. To ensure fairness, we report the performance of each method under two configurations, where the average number of bits per weight (bpw) is set to 3.25 and 3.5. For SQ methods, we take the scale size into account when calculating the bpw. To achieve 3.25 and 3.5 bits per weight, we set the group size for quantization to 32 and 64 respectively. For VQ methods, we consider not only the bit size occupied by the quantized weights but also the bit size required for storing the codebook to achieve the corresponding bpw. In our method, we dynamically set $\tau_c$ and $\tau_f$ according to different models, ensuring that SQ with a bpw of 3.25 is used in nine-tenths of the layers, while VQ with a bpw of 3.5 is used in one-tenth. For example, in RWKV7, $\tau_c$ is set to 1.54, while $\tau_f$ is set to 30. For both vision and language tasks, we select 128 samples from the corresponding test datasets for calibration.

### 4.2. Overall Results

**Performance Comparison on Language Tasks.** As shown in Table 2, on language tasks our method consistently outperforms other approaches across nearly all models. Compared to methods with a bpw of 3.25, regardless of whether they based on SQ or VQ, our method demonstrates signifi-

Table 3: Comparative results under different quantization settings for Vision RWKV models.

| Bpw. | Method | RWKV-T | | | RWKV-S | | |
| --- | --- | --- | --- | --- | --- | --- | --- |
| | | Cls. | Det. | Seg. | Cls. | Det. | Seg. |
| 16 | FloatingPoint | 75.10 | 41.70 | 43.3 | 80.10 | 44.8 | 47.2 |
| 3.5 | GPTQ | 69.74 | 39.85 | 41.20 | 78.30 | 43.37 | 45.50 |
| | AWQ | 68.50 | 39.03 | 38.88 | 78.00 | 42.90 | 42.88 |
| | GPTVQ | 70.31 | 40.14 | 41.65 | 78.65 | **44.03** | 45.00 |
| | VPTQ | 67.21 | 39.02 | 40.14 | 76.40 | 42.01 | 43.54 |
| 3.275 | **Ours** | **70.41** | **40.22** | **41.70** | **78.74** | 43.95 | **46.09** |

cant improvements in both PPL and accuracy on zero-shot tasks. Compared to methods with a bpw of 3.5, our method consistently achieves lower PPL. Except for slightly lower accuracy on RWKV7-0.5B and RWKV6-1B with certain methods, it achieves the highest accuracy across all other models. It can be observed that on the smallest 0.1B model, the PPL of other methods increases by at least 10 points, whereas our method results in an increase of only 4.2 points. On larger models such as RWKV6-7B and RWKV7-14B, our method results in almost no increase in PPL, while the accuracy decreases by less than 1 point.

**Performance Comparison on Vision Tasks.** Table 3 presents the results of the quantized RWKV models applied to various vision tasks, including classification, detection and segmentation. Our method achieves the highest scores in both segmentation and classification tasks. For detection tasks, although the precision of RWKV-S is not the highest, it is very close to the best-performing method.

**Memory Occupancy and Computational Cost.** Our method incurs only negligible loss in 3.275-bpw quantization, making 3.275-bpw inference feasible. As described in the section 1, models based on the RWKV architecture differ from those built on GPT or LLaMA architectures. Whether in the pre-fill or decoder stage, RWKV models exhibit a lower compute-to-memory-access ratio. Consequently, quantizing the weights to lower bit-widths can significantly reduce memory access time, thereby accelerating

Table 4: Comparison of generation speed and memory usage before and after 3.275-bpw quantization on RWKV6 models. All tests were conducted on an NVIDIA A6000 GPU.

| Model Size | speed (tokens/sec) | | | Memory use (GB) | | |
| --- | --- | --- | --- | --- | --- | --- |
| | FP | Quantized | Speed up | FP | Quantized | Mem. saving |
| 3B | 32.95 | 51.29 | 1.55x | 5.88 | 1.65 | 3.56x |
| 7B | 30.75 | 62.42 | 2.03x | 13.91 | 4.25 | 3.27x |
| 14B | 16.02 | 34.32 | 2.14x | 26.07 | 9.21 | 2.83x |

Table 5: Ablation study on the impact of hybrid quantization on LAMBADA PPL and zero-shot[9] score for language RWKV models.

| Model | GPTQ | | GPTVQ | | Ours | |
| --- | --- | --- | --- | --- | --- | --- |
| | 0-shot[9] | LambA. | 0-shot[9] | LambA. | 0-shot[9] | LambA. |
| | Avg.(↑) | (↓) | Avg.(↑) | (↓) | Avg.(↑) | (↓) |
| RWKV7-0.1B | 39.33 | 40.16 | 38.49 | 55.30 | 40.69 | 24.71 |
| RWKV7-0.5B | 45.36 | 13.07 | 43.85 | 20.16 | 45.03 | 13.49 |
| RWKV7-1.47B | 52.28 | 6.55 | 51.31 | 6.85 | 52.23 | 6.54 |
| RWKV6-1B | 51.20 | 5.86 | 49.70 | 5.52 | 51.44 | 5.32 |
| RWKV6-3B | 55.24 | 4.54 | 54.86 | 4.41 | 55.40 | 3.97 |
| RWKV6-7B | 59.20 | 3.59 | 48.29 | 3.41 | 60.18 | 3.21 |
| RWKV6-14B | 61.14 | 3.29 | 59.86 | 3.31 | 62.03 | 2.89 |

the model's inference speed, as shown in Table 4.

### 4.3. Ablation Study

**Hybrid Quantization.** We conduct a series of ablation studies on the hybrid quantization method proposed in Section 3.1, comparing its performance on the RWKV model with that of employing single quantization methods. For fairness, the weights of all multiplication operations are quantized using the RTN method. Our method leverages the proposed coarse-grained and fine-grained proxy to hybridize GPTQ and GPTVQ. While GPTQ and GPTVQ use a bpw of 3.5, our method achieves a bpw of 3.275 by applying GPTVQ (bpw 3.5) to one-tenth of the layers and GPTQ (bpw 3.25) to the remaining nine-tenths. The ablation study results in Table 5 highlight the effectiveness of the hybrid quantization method. In nearly all RWKV models, the hybrid method achieves better metrics compared to both GPTQ and GPTVQ.

**Proxy Strategy.** Table 5 shows that our hybrid quantization improves accuracy but still lags behind floating-point precision. We then apply the proxy described in Section 3.1, combining coarse-grained and fine-grained proxies to determine the quantization method for each layer. The ablation results,

Table 6: Ablation study on the impact of different proxies for hybrid quantization in language RWKV models.

| Method | RWK7-0.1B | | RWK7-0.5B | | RWK7-1.47B | |
| --- | --- | --- | --- | --- | --- | --- |
| | 0-shot[9] | LambA. | 0-shot[9] | LambA. | 0-shot[9] | LambA. |
| | Avg.(↑) | (↓) | Avg.(↑) | (↓) | Avg.(↑) | (↓) |
| Variance | 40.67 | 20.80 | 42.51 | 9.74 | 51.29 | 5.90 |
| CV | 39.09 | 22.36 | 40.92 | 10.10 | 51.58 | 5.79 |
| Range | 39.92 | 23.78 | 40.24 | 10.41 | 51.37 | 5.82 |
| MAD | 38.97 | 22.65 | 42.33 | 10.02 | 51.95 | 6.04 |
| MSE | 37.99 | 28.56 | 42.60 | 10.22 | 51.05 | 6.87 |
| IE | 41.01 | 20.03 | 45.12 | 9.67 | 52.12 | 5.31 |
| **Ours** | **41.04** | **19.70** | **45.54** | **9.55** | **52.32** | **5.24** |

Table 7: Ablation study on the impact of codebook optimization for element-wise multiplication.

| Model | wo. | | w. | |
|---|---|---|---|---|
| | 0-shot[9] Avg.($\uparrow$) | Lambda ($\downarrow$) | 0-shot[9] Avg.($\uparrow$) | Lambda ($\downarrow$) |
| RWKV7-0.1B | 40.69 | 24.71 | 41.09 | 18.41 |
| RWKV7-0.5B | 45.03 | 13.49 | 46.01 | 9.39 |
| RWKV7-1.47B | 52.24 | 6.54 | 52.41 | 5.24 |
| RWKV6-1B | 51.45 | 5.32 | 51.69 | 5.29 |
| RWKV6-3B | 55.48 | 3.97 | 55.79 | 3.88 |
| RWKV6-7B | 60.18 | 3.21 | 60.19 | 3.23 |
| RWKV6-14B | 62.03 | 2.89 | 62.69 | 2.89 |

comparing the use of different proxies such as Variance, Coefficient of Variation (CV) (Abdi, 2010), Range (Gumbel, 1947), Mean Absolute Deviation (MAD) (Konno & Koshizuka, 2005), Mean Squared Error(MSE), and IE, are presented in Table 6. Notably, MSE denotes making selections between SQ and VQ by directly comparing their MSE of each weight. The other metrics are used in the same manner as described in our method, focusing on the transformed weights $G'$. Intuitively, the MSE method is the local optimum for each weight. However, our coarse-to-fine proxy attains the best results across all three models from the global perspective.

**Codebook Optimization.** We conduct ablation experiments on the codebook optimization for the element-wise multiplication proposed in Section 3.2, across all RWKV models. The results are presented in Table 7. It can be observed that using the codebook optimization for the element-wise multiplication operator generates better accuracy across all models compared to not applying the optimization. Specifically, we also visualize the effectiveness of the clipping-based percentile technique within this codebook optimization. From Figure 4, it can be clearly observed that the input activation approximately follows the normal distribution. However, the outliers make the representative feature to leave far from the center point, thereby decreasing the overall performance. By clipping these outliers, a more close-to-center feature can be obtained, thus enhancing the calibration process.

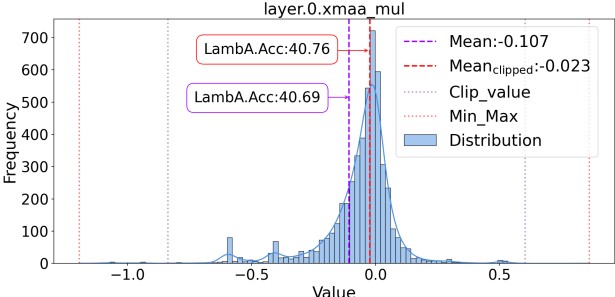

Figure 4: Effectiveness of clipping for batch integration.

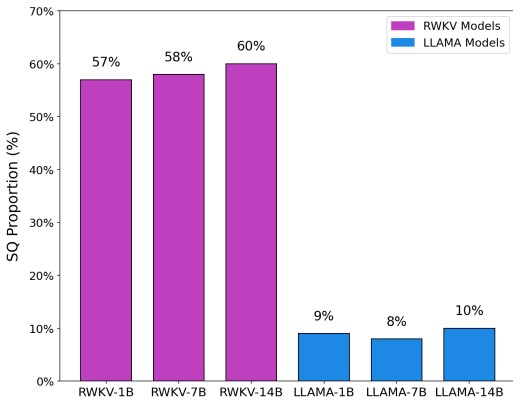

Figure 5: Comparison of SQ proportion between RWKV and LLaMA Models.

### 4.4. More Uniform Weights in RWKV

Table 1 in Section 1 presents the average relative clustering loss of weights using K-Means methods (Lloyd, 1982) for the RWKV family and the LLaMA family respectively. In depth, we conduct experiments leveraging the proposed coarse-to-fine proxy in Section 3.1 to investigate the usage proportions of SQ and VQ. Under the settings of $\tau_c = 1.5$ and $\tau_f = 50$, Figure 5 shows that approximately 60% of the layers in the RWKV family are categorized as suitable for scaler quantization, whereas the proportion is only about 10% for the LLaMA family. This further demonstrates that the RWKV models have a significantly higher number of uniform weights.

## 5. Conclusion

In this paper, we focus on introducing the quantization techniques into the realm of RWKV models. Our investigation reveals that applying SQ or VQ individually may not be optimal for RWKV. We have subsequently identified that enhancing VQ with conventional compensation-based SQ holds great promise. To this end, we propose RWKVQuant, a comprehensive post training quantization framework especially designed for RWKV models. The core idea is to design an optimal strategy that indicates the choice between SQ and VQ for each weight. Specifically, we propose a guidance that employs a coarse-grained proxy to evaluate uniformity and a fine-grained proxy to identify outliers. We also optimize the codebook generation for element-wise multiplication modules, which are unique to the RWKV models. Our proposed RWKVQuant advances in accuracy for both RWKV-based vision and language tasks compared to existing methods, making RWKV models more practical for deployment in resource-constrained environments. As a pioneering study on quantization within the RWKV family, we will publish the code in the hope of promoting further research and facilitating advancements in this field.

## Impact Statement

This paper aims to promote the application of the RWKV family, mainly focused on the post training quantization methods. By introducing RWKVQuant, our approach enables the deployment of RWKV models on resource-constrained devices.

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

# A. Appendix

### A.1. Structure of Time- and Channel-Mixing

The RWKV model, similar to Transformer networks, is composed of multiple identical blocks, each containing a Time Mixing component. The Time Mixing process can be written as:

$$\boldsymbol{r}_t = \boldsymbol{W}_r \cdot (\boldsymbol{\mu}_r \odot \boldsymbol{x}_t + (1 - \boldsymbol{\mu}_r) \odot \boldsymbol{x}_{t-1}), \tag{20}$$

$$\boldsymbol{k}_t = \boldsymbol{W}_k \cdot (\boldsymbol{\mu}_k \odot \boldsymbol{x}_t + (1 - \boldsymbol{\mu}_k) \odot \boldsymbol{x}_{t-1}), \tag{21}$$

$$\boldsymbol{v}_t = \boldsymbol{W}_v \cdot (\boldsymbol{\mu}_v \odot \boldsymbol{x}_t + (1 - \boldsymbol{\mu}_v) \odot \boldsymbol{x}_{t-1}), \tag{22}$$

$$\boldsymbol{wkv}_t = \frac{\sum_{i=1}^{t-1} f(i) \odot \boldsymbol{v}_i + \exp(\boldsymbol{u} + \boldsymbol{k}_t) \odot \boldsymbol{v}_t}{\sum_{i=1}^{t-1} f(i) + \exp(\boldsymbol{u} + \boldsymbol{k}_t)} \tag{23}$$
$$where f(i) = \exp(-(t-1-i)\boldsymbol{w} + \boldsymbol{k}_i),$$

$$\boldsymbol{o}_t = \boldsymbol{W}_o \cdot (\sigma(\boldsymbol{r}_t) \odot \boldsymbol{wkv}_t). \tag{24}$$

Here, the symbol '$\odot$'represents element-wise multiplication, while the symbol ' $\cdot$ 'stands for matrix multiplication. Both $W$ and $\mu$ are parameters. In the context of RWKV, the terms $r_t$, $k_t$, and $v_t$ bear an analogy to the Q, K, and V components found in the attention mechanism of Transformers. Notably, the input x in RWKV is not simply the embedding of the current token. Rather, it signifies the weighted sum of the embedding of the current token and that of the previous token. Subsequently, the Channel Mixing module performs:

$$\boldsymbol{r}'_t = \boldsymbol{W}'_r \cdot (\boldsymbol{\mu}'_r \odot \boldsymbol{x}_t + (1 - \boldsymbol{\mu}'_r) \odot \boldsymbol{x}_{t-1}), \tag{25}$$

$$\boldsymbol{k}'_t = \boldsymbol{W}'_k \cdot (\boldsymbol{\mu}'_k \odot \boldsymbol{x}_t + (1 - \boldsymbol{\mu}'_k) \odot \boldsymbol{x}_{t-1}), \tag{26}$$

$$\boldsymbol{o}'_t = \sigma(\boldsymbol{r}'_t) \odot (\boldsymbol{W}'_v \cdot \max(\boldsymbol{k}'_t, 0)^2). \tag{27}$$

### A.2. RWKV Weight Distribution

Figue 6 shows layers with relatively uniform weight distributions in the RWKV7-0.1B model, which are classified as layers that should use SQ based on our proposed coarse-to-fine proxy. In contrast, Figure 7 illustrates layers with uneven weight distributions, which are typically classified as layers that should use VQ. Furthermore, although the weights in Figure 8 appear generally uniform, the presence of local unevenness still leads to their classification as layers that require VQ.



Figure 6: Unifrom weights without outliers in RWKV7-0.1B different layers.



Figure 7: Non-uniform weights in RWKV7-0.1B different layers.



Figure 8: Unifrom weights with outliers in RWKV7-0.1B different layers.

## A.3. Compute-to-memory Ratio

The figure 9 compares the Compute-to-Memory Ratio (FLOPs per byte) across various models and highlights that RWKV consistently exhibits the lowest ratio, indicating that its operations rely more on memory access rather than intensive computations, compared to models like GPT-3 and LLAMA. This characteristic makes RWKV particularly well-suited for acceleration through weight quantization, as its lower computational demands relative to memory usage allow for more significant gains in inference speed, especially when optimizing for resource-constrained environments.

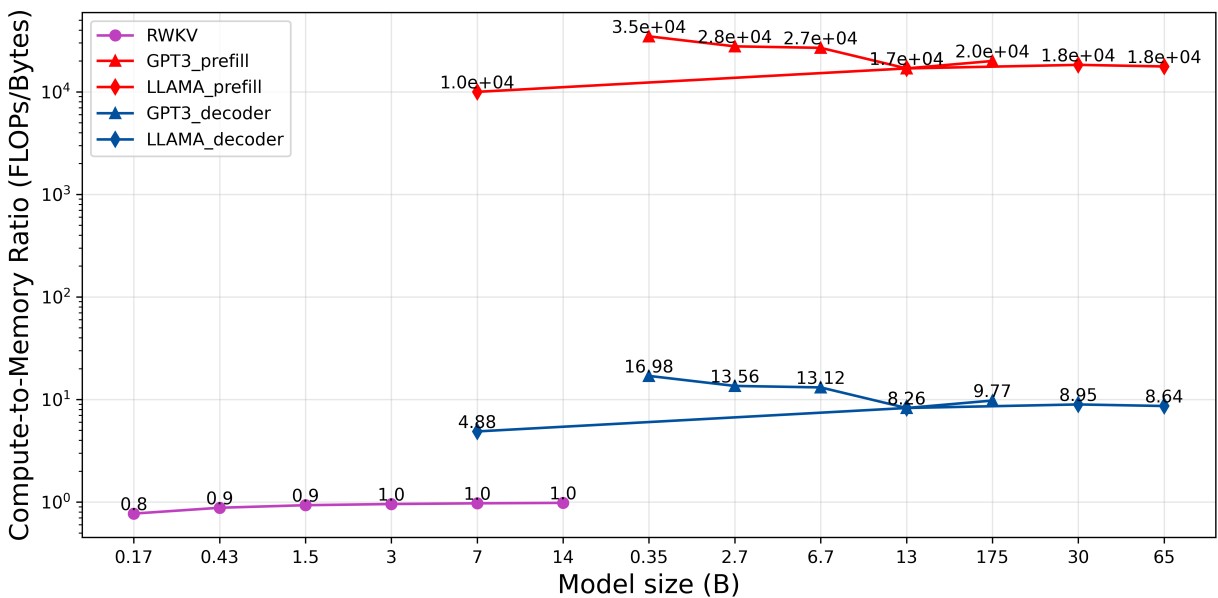

Figure 9: Compute-to-memory-ratio for different models.

Table 8: Complete comparative results under different quantization settings for Vision RWKV models. For classification tasks, we report the Top-1 Accuracy on ImageNet. For detection tasks, the Box AP is evaluated on Coco, while for segmentation tasks, the mIoU is measured on ADE20K.

| Bpw. | Method | RWKV-T | | | RWKV-S | | | RWKV-B | | | RWKV-L | | | RWKV6-T | | | RWKV-L | | | RWKV6-T | | |
|---|---|---|---|---|---|---|---|---|---|---|---|---|---|---|---|---|---|---|---|---|---|---|---|
| | | Cls. | Det. | Seg. | Cls. | Det. | Seg. | Cls. | Det. | Seg. | Cls. | Det. | Seg. | Cls. | Det. | Seg. | Cls. | Det. | Seg. | Cls. | Det. | Seg. |
| 16 | FloatingPoint | 75.10 | 41.70 | 43.30 | 80.10 | 44.80 | 47.20 | 82.00 | 46.80 | 49.20 | - | 50.6 | 53.50 | 76.60 | - | - | 81.10 | - | - | 82.60 | - | - |
| 3.5 | GPTQ | 69.74 | 39.85 | 41.20 | 78.30 | 43.37 | 45.50 | 81.42 | 46.14 | 48.64 | - | 50.30 | 53.26 | 72.79 | - | - | 80.13 | - | - | 82.31 | - | - |
| | AWQ | 68.50 | 39.03 | 38.88 | 78.00 | 42.90 | 42.88 | 81.15 | 45.70 | 48.55 | - | 50.19 | 53.18 | 71.46 | - | - | 79.74 | - | - | 82.09 | - | - |
| | GPTVQ | 70.61 | 40.14 | 41.65 | 78.65 | 44.03 | 45.00 | 81.37 | 46.23 | 48.70 | - | 50.28 | 52.90 | 73.22 | - | - | 80.23 | - | - | 82.25 | - | - |
| | VPTQ | 67.21 | 39.02 | 40.14 | 76.40 | 42.01 | 43.54 | 80.29 | 45.02 | 48.68 | - | 49.10 | 51.45 | 70.36 | - | - | 77.75 | - | - | 81.31 | - | - |
| 3.275 | **Ours** | 70.41 | 40.22 | 41.70 | 78.74 | 43.85 | 46.09 | 81.58 | 46.40 | 48.49 | - | 50.31 | 52.95 | 73.13 | - | - | 80.24 | - | - | 82.32 | - | - |

## A.4. Additional Results

We provide a comprehensive presentation of our results across various datasets to complement the main paper. Specifically, the results include:

- Complete comparison of the results under different quantization settings for Vision RWKV models.(Table 8).

- Complete comparison of the perplexity score on LAMBADA and averaged accuracy on zero-shot common sense reasoning tasks on RWKV7 (Tab 9) and RWKV6 (Table 10).

- Validate the effectiveness of codebook optimization for element-wise multiplication. (Table 11).

Table 9: Complete comparison of perplexity on LAMBADA and averaged accuracy on nine Zero-Shot tasks. For all methods except ours and floating-point, we report metrics under both bpw settings of 3.5 and 3.25.

| Model | Bpw. | Method | ARC-c (↑) | ARC-e (↑) | HQA. (↑) | HellaS. (↑) | Lam. (↑) | OBQA (↑) | PIQA (↑) | SCIQ (↑) | WinoG. (↑) | Avg. (↑) | Wiki2 (↓) |
|---|---|---|---|---|---|---|---|---|---|---|---|---|---|
| RWKV-7 0.1B | 16 | Full Precision | 19.7 | 47.90 | 25.12 | 31.59 | 45.62 | 17.2 | 65.61 | 81.8 | 52.6 | 43.02 | 14.21 |
| | 3.25 | RTN | 19.53 | 40.66 | 23.70 | 30.12 | 12.83 | 14.4 | 62.73 | 70.2 | 51.93 | 36.22 | 152.82 |
| | | GPTQ | 20.39 | 42.17 | 23.52 | 30.59 | 20.47 | 15.40 | 62.62 | 73.60 | 52.50 | 37.92 | 63.54 |
| | | AWQ | 20.13 | 41.16 | 22.72 | 29.88 | 15.20 | 25.44 | 61.15 | 76.70 | 52.40 | 38.31 | 132.06 |
| | | QuaRot | 19.96 | 39.94 | 22.46 | 29.38 | 12.28 | 13.80 | 60.93 | 58.80 | 53.19 | 34.53 | 243.99 |
| | | KMeans | 18.94 | 43.56 | 23.81 | 29.85 | 19.93 | 15.00 | 63.49 | 75.70 | 53.59 | 38.21 | 87.06 |
| | | GPTVQ | 18.94 | 43.93 | 22.75 | 30.04 | 38.06 | 16.40 | 63.36 | 77.00 | 51.77 | 40.25 | 23.75 |
| | | VPTQ | 18.94 | 39.60 | 21.84 | 28.59 | 17.50 | 13.20 | 59.90 | 71.90 | 50.51 | 35.78 | 128.59 |
| | 3.5 | RTN | 19.36 | 43.47 | 24.50 | 30.17 | 18.13 | 16.20 | 64.09 | 74.40 | 52.48 | 38.08 | 81.14 |
| | | GPTQ | 18.68 | 44.06 | 24.14 | 30.69 | 26.74 | 17.80 | 63.76 | 74.30 | 53.82 | 39.33 | 40.16 |
| | | AWQ | 19.88 | 40.40 | 22.97 | 29.52 | 14.92 | 14.00 | 59.99 | 71.50 | 52.64 | 36.20 | 55.72 |
| | | QuaRot | 19.88 | 44.40 | 22.72 | 30.16 | 15.36 | 15.60 | 62.78 | 72.40 | 52.09 | 37.26 | 126.19 |
| | | KMeans | 19.19 | 44.02 | 24.14 | 29.98 | 29.44 | 15.00 | 63.54 | 77.40 | 53.27 | 39.55 | 36.23 |
| | | GPTVQ | 19.11 | 42.88 | 22.68 | 30.25 | 38.35 | 15.80 | 63.11 | 74.40 | 54.30 | 40.10 | 25.82 |
| | | VPTQ | 18.51 | 40.74 | 22.42 | 28.82 | 23.17 | 13.80 | 61.31 | 72.90 | 51.93 | 37.07 | 74.70 |
| | 3.275 | **Ours** | 19.73 | 42.17 | 24.14 | 30.02 | 42.25 | 17.00 | 63.10 | 79.10 | 52.30 | 41.10 | 18.41 |
| RWKV-7 0.5B | 16 | Full Precision | 23.80 | 57.11 | 28.55 | 38.28 | 57.85 | 20.80 | 69.31 | 86.6 | 55.72 | 48.67 | 7.21 |
| | 3.25 | RTN | 21.84 | 45.28 | 23.66 | 32.64 | 26.45 | 19.20 | 64.96 | 73.4 | 52.56 | 39.99 | 57.11 |
| | | GPTQ | 22.61 | 52.77 | 26.14 | 35.92 | 42.52 | 20.80 | 67.46 | 79.60 | 55.40 | 43.69 | 15.97 |
| | | AWQ | 19.62 | 43.09 | 22.93 | 32.64 | 39.34 | 17.00 | 62.57 | 79.20 | 54.14 | 41.16 | 23.29 |
| | | QuaRot | 22.18 | 48.69 | 24.76 | 33.85 | 20.22 | 19.00 | 65.01 | 73.80 | 54.06 | 40.17 | 76.89 |
| | | KMeans | 23.46 | 52.48 | 26.29 | 36.19 | 39.41 | 20.00 | 67.35 | 81.50 | 54.69 | 44.59 | 20.19 |
| | | GPTVQ | 21.50 | 51.30 | 26.14 | 34.80 | 46.19 | 19.40 | 66.64 | 82.40 | 54.38 | 43.64 | 14.15 |
| | | VPTQ | 18.68 | 42.97 | 22.86 | 31.15 | 34.81 | 16.40 | 62.73 | 78.80 | 52.88 | 40.14 | 30.63 |
| | 3.5 | RTN | 18.13 | 19.36 | 43.47 | 24.50 | 30.17 | 16.20 | 64.09 | 74.40 | 52.48 | 38.08 | 81.14 |
| | | GPTQ | 22.26 | 53.11 | 25.41 | 36.51 | 45.78 | 21.40 | 67.51 | 83.10 | 56.51 | 45.73 | 13.07 |
| | | AWQ | 21.33 | 43.60 | 22.57 | 33.06 | 45.14 | 18.80 | 61.75 | 82.60 | 52.80 | 42.40 | 16.98 |
| | | QuaRot | 28.33 | 20.13 | 46.38 | 24.54 | 32.30 | 18.80 | 65.23 | 80.40 | 51.46 | 40.84 | 40.38 |
| | | KMeans | 20.73 | 47.55 | 25.05 | 33.58 | 41.39 | 17.40 | 65.83 | 82.00 | 54.14 | 43.07 | 17.05 |
| | | GPTVQ | 21.92 | 52.18 | 25.38 | 34.90 | 52.05 | 17.80 | 67.51 | 81.60 | 53.82 | 44.13 | 10.88 |
| | | VPTQ | 19.79 | 44.73 | 23.12 | 31.62 | 37.78 | 16.40 | 63.98 | 78.90 | 53.27 | 41.06 | 25.03 |
| | 3.275 | **Ours** | 22.61 | 53.11 | 25.41 | 34.45 | 53.39 | 19.60 | 66.53 | 84.20 | 54.80 | 46.01 | 9.39 |
| RWKV-7 1.47B | 16 | Full Precision | 31.65 | 65.40 | 32.12 | 46.55 | 66.97 | 26.00 | 72.68 | 90.00 | 64.40 | 55.08 | 4.84 |
| | 3.25 | RTN | 21.69 | 53.44 | 25.65 | 35.81 | 46.63 | 17.90 | 65.39 | 85.14 | 57.55 | 45.46 | 11.43 |
| | | GPTQ | 29.01 | 59.93 | 28.99 | 42.74 | 55.52 | 23.60 | 70.34 | 87.30 | 62.98 | 51.15 | 7.93 |
| | | AWQ | 22.35 | 44.91 | 22.72 | 35.22 | 45.04 | 16.00 | 61.26 | 85.50 | 59.58 | 43.62 | 15.27 |
| | | QuaRot | 29.43 | 63.04 | 30.01 | 42.24 | 52.30 | 22.80 | 70.78 | 88.10 | 58.64 | 50.81 | 9.39 |
| | | KMeans | 29.86 | 60.39 | 28.92 | 43.07 | 57.75 | 23.20 | 70.78 | 90.20 | 62.90 | 51.89 | 7.04 |
| | | GPTVQ | 27.21 | 61.48 | 28.51 | 42.95 | 64.84 | 22.80 | 70.34 | 88.40 | 62.03 | 52.06 | 5.54 |
| | | VPTQ | 22.78 | 51.55 | 25.23 | 36.13 | 49.31 | 18.20 | 66.21 | 85.40 | 57.69 | 45.83 | 11.13 |
| | 3.5 | RTN | 29.94 | 61.82 | 30.45 | 43.81 | 54.84 | 24.2 | 71.27 | 88.1 | 61.32 | 51.74 | 7.89 |
| | | GPTQ | 30.54 | 61.65 | 29.79 | 43.96 | 59.23 | 23.80 | 71.65 | 88.90 | 61.06 | 52.28 | 6.55 |
| | | AWQ | 22.44 | 46.00 | 21.95 | 35.30 | 52.65 | 17.20 | 61.86 | 84.50 | 59.66 | 44.61 | 10.71 |
| | | QuaRot | 30.80 | 62.28 | 30.85 | 43.27 | 55.64 | 23.20 | 71.87 | 89.20 | 60.77 | 51.98 | 7.94 |
| | | KMeans | 29.69 | 61.44 | 30.48 | 43.92 | 61.58 | 24.60 | 71.32 | 89.90 | 62.03 | 52.77 | 6.57 |
| | | GPTVQ | 28.32 | 61.57 | 29.24 | 42.63 | 64.18 | 21.20 | 70.83 | 88.60 | 62.66 | 52.13 | 5.51 |
| | | VPTQ | 23.54 | 54.25 | 26.44 | 37.70 | 52.24 | 19.04 | 67.73 | 86.90 | 58.64 | 47.38 | 9.52 |
| | 3.275 | **Ours** | 28.15 | 61.32 | 29.35 | 43.24 | 65.38 | 22.60 | 71.70 | 88.90 | 61.01 | 52.40 | 5.24 |

Table 10: Complete comparison of perplexity on LAMBADA and averaged accuracy on nine Zero-Shot tasks. For all methods except ours and floating-point, we report metrics under both bpw settings of 3.5 and 3.25.

| Model | Bpw. | Method | ARC-c (↑) | ARC-e (↑) | HQA. (↑) | HellaS. (↑) | Lam. (↑) | OBQA (↑) | PIQA (↑) | SCIQ (↑) | WinoG. (↑) | Avg. (↑) | Wiki2 (↓) |
|---|---|---|---|---|---|---|---|---|---|---|---|---|---|
| RWKV-6 1B | 16 | Full Precision | 31.22 | 64.40 | 30.45 | 46.34 | 67.11 | 25.20 | 74.26 | 89.60 | 60.93 | 54.39 | 4.60 |
| | 3.25 | RTN | 25.25 | 55.43 | 27.13 | 42.59 | 59.73 | 24.00 | 71.00 | 84.80 | 58.64 | 49.84 | 6.39 |
| | | GPTQ | 25.17 | 58.62 | 27.06 | 42.07 | 60.18 | 24.00 | 71.16 | 86.70 | 60.06 | 50.55 | 6.43 |
| | | AWQ | 23.72 | 50.16 | 23.12 | 37.34 | 40.03 | 19.60 | 65.12 | 83.50 | 57.14 | 44.41 | 17.97 |
| | | QuaRot | 27.30 | 51.89 | 26.65 | 37.42 | 32.08 | 21.20 | 67.84 | 85.00 | 55.80 | 45.02 | 29.38 |
| | | KMeans | 27.98 | 55.21 | 26.80 | 40.03 | 41.24 | 20.20 | 69.53 | 85.20 | 57.06 | 47.02 | 15.93 |
| | | GPTVQ | 24.65 | 57.82 | 27.20 | 39.67 | 62.33 | 22.00 | 70.02 | 86.30 | 58.80 | 49.86 | 6.11 |
| | | VPTQ | 21.33 | 49.45 | 23.59 | 34.65 | 42.95 | 17.80 | 65.56 | 84.40 | 55.32 | 43.89 | 14.67 |
| | 3.5 | RTN | 29.43 | 60.81 | 27.46 | 43.64 | 61.11 | 22.60 | 72.52 | 88.40 | 59.90 | 51.76 | 5.83 |
| | | GPTQ | 26.53 | 59.55 | 27.53 | 43.63 | 61.61 | 23 | 71.76 | 87.9 | 59.58 | 51.23 | 5.86 |
| | | AWQ | 24.57 | 53.07 | 22.90 | 38.57 | 50.48 | 20.40 | 66.43 | 85.20 | 58.87 | 51.20 | 10.97 |
| | | QuaRot | 26.19 | 54.67 | 26.91 | 39.21 | 41.34 | 23.60 | 68.60 | 85.70 | 56.82 | 47.00 | 16.29 |
| | | KMeans | 28.07 | 60.26 | 28.11 | 41.84 | 51.50 | 22.40 | 70.23 | 87.80 | 60.14 | 50.03 | 8.24 |
| | | GPTVQ | 27.81 | 58.71 | 26.18 | 40.41 | 61.96 | 22.20 | 70.83 | 87.30 | 57.22 | 50.29 | 5.74 |
| | | VPTQ | 20.87 | 49.99 | 23.41 | 34.37 | 43.41 | 17.80 | 64.20 | 85.50 | 54.85 | 43.82 | 14.74 |
| | 3.275 | **Ours** | 27.90 | 60.35 | 28.08 | 42.51 | 63.87 | 23.00 | 71.38 | 87.90 | 60.22 | 51.69 | 5.29 |
| RWKV-6 3B | 16 | Full Precision | 35.58 | 71.33 | 33.29 | 50.53 | 71.32 | 28.00 | 76.15 | 92.30 | 66.45 | 58.32 | 3.83 |
| | 3.25 | RTN | 30.54 | 64.52 | 28.55 | 47.21 | 66.87 | 24.8 | 73.34 | 88.8 | 62.98 | 54.17 | 4.71 |
| | | GPTQ | 31.14 | 65.06 | 28.77 | 46.92 | 65.65 | 25.40 | 73.83 | 85.1 | 63.61 | 53.94 | 4.88 |
| | | AWQ | 26.02 | 55.13 | 23.48 | 40.42 | 47.53 | 17.00 | 66.05 | 87.80 | 61.79 | 47.24 | 11.97 |
| | | QuaRot | 29.43 | 57.82 | 28.55 | 41.59 | 38.35 | 24.00 | 69.74 | 83.80 | 60.93 | 48.24 | 22.67 |
| | | KMeans | 31.99 | 66.41 | 30.99 | 44.62 | 52.26 | 25.60 | 72.41 | 92.10 | 61.24 | 53.06 | 8.27 |
| | | GPTVQ | 29.77 | 64.52 | 28.84 | 44.56 | 69.92 | 24.40 | 72.57 | 91.20 | 62.35 | 54.23 | 4.31 |
| | | VPTQ | 25.00 | 56.56 | 25.20 | 37.93 | 55.95 | 18.40 | 67.57 | 86.20 | 61.24 | 48.22 | 7.77 |
| | 3.5 | RTN | 31.56 | 66.75 | 30.78 | 47.91 | 68.02 | 26.00 | 74.75 | 90.50 | 53.53 | 54.42 | 4.50 |
| | | GPTQ | 31.48 | 67.17 | 29.80 | 47.65 | 67.49 | 25.80 | 74.26 | 90.30 | 63.22 | 55.24 | 4.54 |
| | | AWQ | 28.41 | 54.67 | 24.03 | 39.96 | 53.06 | 19.20 | 66.15 | 88.20 | 61.95 | 48.40 | 8.77 |
| | | QuaRot | 31.22 | 61.48 | 29.46 | 43.04 | 42.99 | 29.20 | 71.21 | 89.10 | 61.48 | 51.01 | 16.99 |
| | | KMeans | 34.72 | 67.46 | 31.43 | 46.63 | 58.53 | 25.60 | 73.77 | 91.90 | 64.95 | 54.99 | 6.28 |
| | | GPTVQ | 30.04 | 65.61 | 29.57 | 45.39 | 69.84 | 24.60 | 73.12 | 89.90 | 65.58 | 54.85 | 4.12 |
| | | VPTQ | 25.59 | 56.81 | 25.52 | 38.09 | 53.72 | 19.60 | 68.49 | 87.70 | 60.69 | 48.46 | 8.62 |
| | 3.275 | **Ours** | 30.97 | 65.74 | 30.12 | 46.90 | 71.18 | 25.00 | 74.53 | 92.20 | 65.51 | 55.79 | 3.88 |
| RWKV-6 7B | 16 | Full Precision | 41.70 | 75.25 | 35.66 | 55.82 | 75.35 | 31.4 | 78.18 | 93.80 | 68.11 | 61.69 | 3.21 |
| | 3.25 | RTN | 35.66 | 70.32 | 32.53 | 51.85 | 70.04 | 28.20 | 77.20 | 92.60 | 66.69 | 58.34 | 3.87 |
| | | GPTQ | 38.56 | 72.51 | 31.69 | 51.98 | 71.78 | 29.40 | 77.14 | 93.50 | 67.00 | 59.28 | 3.72 |
| | | AWQ | 29.01 | 57.40 | 23.92 | 40.72 | 52.44 | 20.60 | 67.3 | 88.10 | 65.91 | 49.48 | 8.33 |
| | | QuaRot | 32.25 | 67.46 | 31.36 | 47.39 | 51.43 | 28.40 | 74.59 | 89.60 | 66.21 | 54.29 | 8.81 |
| | | KMeans | 38.73 | 72.72 | 34.71 | 51.41 | 61.85 | 29.00 | 76.93 | 81.50 | 54.69 | 55.72 | 4.69 |
| | | GPTVQ | 36.34 | 71.63 | 32.09 | 49.78 | 74.17 | 28.40 | 74.64 | 92.40 | 67.71 | 58.57 | 3.49 |
| | | VPTQ | 28.83 | 63.80 | 28.18 | 42.15 | 66.23 | 21.20 | 71.32 | 91.10 | 64.79 | 53.06 | 4.75 |
| | 3.5 | RTN | 37.37 | 70.79 | 32.53 | 53.26 | 73.06 | 29.80 | 76.93 | 90.5 | 68.58 | 59.20 | 3.59 |
| | | GPTQ | 39.33 | 73.40 | 32.96 | 52.95 | 71.96 | 30.60 | 77.2 | 94.20 | 68.11 | 60.07 | 3.68 |
| | | AWQ | 29.01 | 57.40 | 23.92 | 40.72 | 61.54 | 20.60 | 67.3 | 88.10 | 65.91 | 50.50 | 7.07 |
| | | QuaRot | 35.32 | 67.76 | 31.36 | 49.68 | 60.18 | 31.40 | 75.46 | 94.50 | 66.92 | 56.95 | 6.44 |
| | | KMeans | 39.84 | 74.03 | 34.35 | 53.01 | 67.48 | 28.60 | 76.98 | 93.90 | 67.48 | 59.51 | 3.96 |
| | | GPTVQ | 38.50 | 72.50 | 31.69 | 51.98 | 75.04 | 29.40 | 77.71 | 93.50 | 67.00 | 59.70 | 3.30 |
| | | VPTQ | 28.78 | 62.33 | 27.83 | 42.45 | 67.86 | 21.00 | 71.49 | 90.40 | 64.48 | 52.95 | 4.47 |
| | 3.275 | **Ours** | 37.62 | 72.93 | 33.63 | 52.03 | 75.70 | 30.40 | 76.60 | 95.10 | 67.71 | 60.19 | 3.23 |
| RWKV-6 14B | 16 | Full Precision | 44.70 | 76.97 | 37.05 | 58.76 | 76.23 | 33.6 | 79.59 | 94.7 | 71.27 | 63.65 | 3.02 |
| | 3.25 | RTN | 39.76 | 73.98 | 32.93 | 55.66 | 74.03 | 31.80 | 79.10 | 93.00 | 70.24 | 61.16 | 3.34 |
| | | GPTQ | 38.22 | 69.99 | 33.36 | 55.28 | 74.00 | 31.40 | 78.18 | 91.50 | 69.69 | 60.18 | 3.43 |
| | | AWQ | 28.15 | 55.34 | 23.77 | 38.82 | 58.59 | 20.60 | 65.61 | 88.90 | 64.40 | 49.35 | 8.18 |
| | | QuaRot | 36.51 | 69.27 | 32.78 | 50.38 | 40.09 | 28.20 | 76.06 | 91.60 | 67.95 | 54.76 | 14.05 |
| | | KMeans | 43.85 | 74.16 | 35.55 | 54.84 | 67.81 | 32.40 | 78.18 | 94.80 | 71.27 | 61.40 | 4.61 |
| | | GPTVQ | 35.65 | 71.66 | 30.45 | 52.33 | 76.46 | 30.14 | 76.31 | 93.24 | 70.44 | 59.63 | 3.15 |
| | | VPTQ | 36.27 | 69.54 | 29.46 | 50.08 | 73.01 | 27.35 | 76.38 | 91.40 | 67.39 | 57.87 | 3.62 |
| | 3.5 | RTN | 39.59 | 72.22 | 33.55 | 56.31 | 74.29 | 31.4 | 78.01 | 92.3 | 69.92 | 60.84 | 3.31 |
| | | GPTQ | 38.90 | 72.72 | 34.46 | 56.34 | 75.08 | 31.20 | 78.67 | 91.70 | 71.19 | 61.14 | 3.29 |
| | | AWQ | 29.09 | 57.82 | 23.59 | 39.99 | 64.47 | 20.20 | 67.19 | 90.70 | 64.71 | 50.86 | 6.06 |
| | | QuaRot | 38.05 | 69.06 | 33.80 | 50.96 | 44.56 | 30.80 | 76.38 | 93.00 | 70.40 | 56.33 | 10.63 |
| | | KMeans | 41.29 | 74.07 | 34.82 | 54.76 | 70.66 | 31.60 | 78.23 | 94.30 | 70.63 | 61.15 | 3.84 |
| | | GPTVQ | 36.65 | 70.87 | 31.91 | 52.43 | 75.04 | 30.00 | 77.31 | 93.40 | 70.24 | 59.76 | 3.34 |
| | | VPTQ | 37.10 | 69.87 | 29.69 | 50.31 | 72.31 | 27.45 | 76.56 | 91.60 | 66.55 | 57.93 | 3.75 |
| | 3.275 | **Ours** | 41.80 | 76.50 | 34.09 | 55.26 | 78.17 | 32.80 | 78.61 | 95.40 | 71.60 | 62.69 | 2.89 |

Table 11: Complete ablation study on the impact of codebook optimization for element-wise multiplication.

| Model | Method | ARC-c (↑) | ARC-e (↑) | HQA. (↑) | HellaS. (↑) | Lam. (↑) | OBQA (↑) | PIQA (↑) | SCIQ (↑) | WinoG. (↑) | Avg. (↑) | Wiki2 (↓) |
|---|---|---|---|---|---|---|---|---|---|---|---|---|
| RWKV7-0.1B | w. | 19.79 | 42.17 | 24.14 | 30.02 | 42.25 | 17.00 | 63.10 | 79.10 | 52.30 | 41.09 | 18.41 |
| | wo. | 20.15 | 44.45 | 24.47 | 30.32 | 38.09 | 17.80 | 64.08 | 73.95 | 52.91 | 40.69 | 24.71 |
| RWKV7-0.5B | w. | 22.61 | 53.11 | 25.41 | 34.45 | 53.39 | 19.60 | 66.53 | 84.20 | 54.80 | 46.01 | 9.39 |
| | wo. | 21.61 | 53.02 | 25.31 | 34.55 | 47.10 | 18.90 | 66.53 | 83.7 | 54.61 | 45.03 | 13.49 |
| RWKV7-1.47B | w. | 28.15 | 61.32 | 29.35 | 43.24 | 65.38 | 22.60 | 71.70 | 88.90 | 61.01 | 52.40 | 5.24 |
| | wo. | 29.15 | 61.90 | 29.99 | 42.92 | 59.31 | 24.10 | 70.13 | 88.52 | 64.14 | 52.24 | 6.54 |
| RWKV6-1B | w. | 27.90 | 60.35 | 28.08 | 42.51 | 63.87 | 23.00 | 71.38 | 87.9 | 60.22 | 51.69 | 5.29 |
| | wo. | 27.38 | 60.05 | 27.42 | 42.31 | 63.71 | 23.2 | 71.72 | 87.5 | 59.75 | 51.44 | 5.32 |
| RWKV6-3B | w. | 30.97 | 65.74 | 30.12 | 46.90 | 71.18 | 25.00 | 74.53 | 92.20 | 65.51 | 55.79 | 3.88 |
| | wo. | 30.97 | 68.00 | 29.72 | 46.84 | 70.46 | 25.20 | 74.21 | 90.80 | 63.14 | 55.48 | 3.97 |
| RWKV6-7B | w. | 37.62 | 72.93 | 33.63 | 52.03 | 75.70 | 30.40 | 76.6 | 95.1 | 67.71 | 60.19 | 3.23 |
| | wo. | 37.97 | 73.19 | 33.44 | 52.04 | 75.18 | 30.80 | 76.61 | 93.8 | 68.59 | 60.18 | 3.21 |
| RWKV6-14B | w. | 41.80 | 76.50 | 34.09 | 55.26 | 78.17 | 32.80 | 78.61 | 95.40 | 71.60 | 62.69 | 2.89 |
| | wo. | 40.53 | 73.82 | 34.13 | 56.19 | 77.72 | 32.00 | 78.44 | 94.00 | 71.50 | 62.03 | 2.89 |

Table 12: Our method's performance on RWKV6, using the LambA training dataset, shows advantages in PPL and ACC metrics under consistent experimental settings.

| | GPTQ | | GPTVQ | | Ours | |
|---|---|---|---|---|---|---|
| | Avg-acc | LambA. PPL | Avg-acc | LambA. PPL | Avg-acc | LambA. PPL |
| RWKV6 - 1B | 51.28 | 5.87 | 49.69 | 5.51 | 51.41 | 5.35 |
| RWKV6 - 3B | 55.25 | 4.56 | 54.85 | 4.42 | 55.38 | 3.97 |
| RWKV6 - 7B | 59.18 | 3.59 | 48.30 | 3.41 | 60.18 | 3.21 |
| RWKV6 - 14B | 61.15 | 3.29 | 59.86 | 3.31 | 62.03 | 2.89 |

Although using the test set for calibration is a common practice in Post - Training Quantization (PTQ), we conducted additional comparative experiments based on the LambA training dataset for RWKV6 in Tab. 12. The results show that our method outperforms both GPTVQ (Vector Quantization) and GPTQ (Scalar Quantization). This verifies the effectiveness and robustness of our approach.

In Equation 17, the hyperparameter $K$ represents the order of Taylor decomposition. Higher-order terms in the Taylor expansion are close to zero and have minimal impact on the final result. In our previous paper, we used $K = 4$ as the configuration. To further validate the influence of $K$ on our method, we conducted comparative experiments on RWKV6 as shown in Tab. 13. Under the same configurations of $\tau_f$ and $\tau_c$, we compared the model accuracy under different values of $K$. It can be observed that when $K$ increases from 2 to 4, the model accuracy significantly improves. However, as $K$ further increases, the accuracy does not continue to increase and even starts to decline. Therefore, we conclude that an excessively large $K$ is unnecessary: it not only fails to improve accuracy but also introduces additional computations.

### A.5. Limitations and Future Work

Our proposed RWKVQuant framework relies on the coarse-to-fine proxy introduced in Section 3.1, where $\tau_c$ and $\tau_f$ play a important role in determining the appropriate quantization method for each layer. In our experiments, these values were empirically set based on different model configurations, ensuring that the 3.25 bpw SQ proportion is approximately one-tenth and nine-tenths for 3.5 bpw VQ across different models.

Actually, as shown in algorithm 1, $\tau_c$ and $\tau_f$ are automatically set for each individual model and do not require adaptation. Specifically, we obtain their values in the following steps:

Table 13: The detailed performance metrics for the RWKV6 model across different $K$, $\tau_f$ and $\tau_c$ configurations. $C$ is the percentile of $M$ (384) values of $P_c$ used to determine $\tau_c$. $F$ is the percentile of $M * C$ values of $P_f$ used to determine $\tau_f$.

| Lambda accuracy SQ:VQ Proportion Quantization Bits | $K = 2$ | $K = 3$ | $K = 4$ | $K = 5$ | $K = 6$ |
|---|---|---|---|---|---|
| $F = 0.2, C = 0.5$ | 63.11 | 63.82 | 63.87 | 63.88 | 63.87 |
| | 10:90 | 10:90 | 10:90 | 10:90 | 10:90 |
| | 3.275 | 3.275 | 3.275 | 3.275 | 3.275 |
| $F = 0.4, C = 0.5$ | 62.95 | 63.71 | 63.78 | 63.74 | 63.78 |
| | 20:80 | 20:80 | 20:80 | 16:84 | 20:80 |
| | 3.300 | 3.300 | 3.300 | 3.290 | 3.300 |
| $F = 0.6, C = 0.5$ | 61.96 | 62.87 | 62.98 | 62.95 | 62.91 |
| | 30:70 | 30:70 | 30:70 | 30:70 | 30:70 |
| | 3.325 | 3.325 | 3.325 | 3.325 | 3.325 |
| $F = 0.8, C = 0.5$ | 60.98 | 61.35 | 61.56 | 61.68 | 61.54 |
| | 40:60 | 40:60 | 40:60 | 40:60 | 40:60 |
| | 3.350 | 3.350 | 3.350 | 3.350 | 3.350 |

1. Compute the coarse-grained proxy $P_c$(Eq.15) for each layer to be quantized.

2. Set $\tau_c$ to the value at the 50th percentile of all $P_c$.

3. Compute the fine-grained proxy $P_f$ (Eq.17) for each layer whose $P_c < \tau_c$.

4. Set $\tau_f$ to the value at the 20th percentile of all $P_f$.

Although fine-tuning the percentile values for each network may further improve accuracy, the percentile values (i.e., 20% and 50%) —used in all our experiments—already delivers strong performance across all RWKV networks, as shown in Table 2.

Table 14: Comparison of perplexity on LAMBADA and averaged accuracy on nine Zero-Shot tasks under different $\tau_c$ and $\tau_f$ configurations for the RWKV models.

| $\tau_c$ | $\tau_f$ | RWKV7-0.1B | | RWKV7-0.5B | | RWKV7-1.47B | |
|---|---|---|---|---|---|---|
| | | 0-shot[9] Avg.($\uparrow$) | LambA. ($\downarrow$) | 0-shot[9] Avg.($\uparrow$) | LambA. ($\downarrow$) | 0-shot[9] Avg.($\uparrow$) | LambA. ($\downarrow$) |
| 1.00 | 20.00 | 40.27 | 23.65 | 43.72 | 13.98 | 52.09 | 5.52 |
| | 25.00 | 40.27 | 23.65 | 43.72 | 13.98 | 52.09 | 5.52 |
| | 30.00 | 40.27 | 23.65 | 43.62 | 14.06 | 52.09 | 5.52 |
| | 35.00 | 40.25 | 23.75 | 43.64 | 14.15 | 52.06 | 5.54 |
| | 40.00 | 40.25 | 23.75 | 43.64 | 14.15 | 52.06 | 5.54 |
| 1.50 | 20.00 | 40.56 | 18.68 | 45.93 | 9.88 | 52.37 | 5.24 |
| | 25.00 | 41.01 | 18.31 | 46.01 | 9.46 | 52.40 | 5.24 |
| | 30.00 | 41.10 | 18.41 | 46.01 | 9.39 | 52.40 | 5.24 |
| | 35.00 | 40.86 | 18.71 | 46.01 | 9.45 | 52.40 | 5.24 |
| | 40.00 | 40.78 | 18.41 | 45.88 | 9.67 | 52.34 | 5.24 |
| 2.00 | 20.00 | 39.86 | 28.12 | 45.93 | 9.88 | 52.34 | 6.12 |
| | 25.00 | 39.86 | 28.12 | 45.93 | 9.88 | 52.34 | 6.12 |
| | 30.00 | 39.65 | 31.34 | 45.87 | 10.07 | 52.26 | 6.32 |
| | 35.00 | 39.37 | 37.25 | 45.83 | 13.01 | 52.28 | 6.54 |
| | 40.00 | 39.37 | 37.25 | 45.83 | 13.01 | 52.28 | 6.54 |

Table 15: The detailed performance metrics for the RWKV6 model under various configurations of the parameters $\tau_f$ and $\tau_c$. $C$ is the percentile of $M$ (384) values of $P_c$ used to determine $\tau_c$. $F$ is the percentile of $M * C$ values of $P_f$ used to determine $\tau_f$.

| Lambda accuracy SQ:VQ Proportion Quantization Bits | $C=0.1$ | $C=0.2$ | $C=0.3$ | $C=0.4$ | $C=0.5$ | $C=0.6$ | $C=0.7$ | $C=0.8$ |
|---|---|---|---|---|---|---|---|---|
| | 61.20 | 62.14 | 61.78 | 61.65 | 61.21 | 60.84 | 60.75 | 60.54 |
| wo F ( wo $\tau_f$ ) | 10:90 | 20:80 | 30:70 | 40:60 | 50:50 | 60:40 | 70:30 | 80:20 |
| | 3.275 | 3.300 | 3.325 | 3.350 | 3.375 | 3.400 | 3.425 | 3.450 |
| | 61.28 | 61.31 | 61.54 | 62.45 | 62.44 | 62.47 | 61.25 | 60.84 |
| $F=0.1$ | 1:99 | 2:98 | 3:97 | 4:96 | 5:95 | 6:94 | 7:93 | 8:92 |
| | 3.253 | 3.255 | 3.258 | 3.260 | 3.263 | 3.265 | 3.268 | 3.270 |
| | 61.29 | 61.52 | 61.88 | 62.63 | **63.87** | **63.88** | 61.34 | 60.88 |
| $F=0.2$ | 2:98 | 4:96 | 6:94 | 8:92 | **10:90** | 12:88 | 14:86 | 16:84 |
| | 3.255 | 3.260 | 3.265 | 3.270 | **3.275** | 3.280 | 3.285 | 3.290 |
| | 61.39 | 61.64 | 62.62 | 62.73 | 63.83 | 62.86 | 60.95 | 60.89 |
| $F=0.3$ | 3:97 | 6:94 | 9:91 | 12:88 | 15:85 | 18:82 | 21:79 | 24:76 |
| | 3.258 | 3.265 | 3.273 | 3.280 | 3.288 | 3.295 | 3.302 | 3.310 |
| | 61.42 | 61.94 | 62.65 | 62.21 | 62.78 | 62.32 | 61.36 | 61.26 |
| $F=0.4$ | 4:96 | 8:92 | 12:88 | 16:84 | 20:80 | 24:76 | 28:72 | 32:68 |
| | 3.260 | 3.270 | 3.280 | 3.290 | 3.300 | 3.310 | 3.320 | 3.330 |
| | 61.43 | 62.37 | 62.54 | 61.95 | 61.18 | 60.92 | 60.73 | 60.54 |
| $F=0.6$ | 6:94 | 12:88 | 18:82 | 24:76 | 30:70 | 36:64 | 42:58 | 48:52 |
| | 3.265 | 3.280 | 3.295 | 3.310 | 3.325 | 3.340 | 3.355 | 3.370 |
| | 61.44 | 61.47 | 61.13 | 60.88 | 60.56 | 60.53 | 60.33 | 60.34 |
| $F=0.8$ | 8:92 | 16:84 | 24:76 | 32:68 | 40:60 | 48:52 | 56:44 | 64:36 |
| | 3.270 | 3.290 | 3.310 | 3.330 | 3.390 | 3.410 | 3.430 | 3.450 |

However, this allocation might not reflect the most balanced or effective proportion. As shown in Table 14, we conducted multiple comparative experiments on the RWKV language model with varying $\tau_c$ and $\tau_f$. The results demonstrate that when $\tau_c$ is larger than the optimal value, the final accuracy approaches that of directly using uniform quantization, whereas a smaller $\tau_c$ yields results closer to codebook quantization. Moreover, with an appropriately chosen $\tau_c$, the setting of $\tau_f$ has a direct impact on the final accuracy.Therefore, in future work, we will further explore how to determine appropriate values for $\tau_c$ and $\tau_f$:

- Based on proper initialization, we plan to utilize fine-tuning to achieve the optimal configuration for different models.

- At the same time, we will remove the fixed constraint of bpw being 3.275 and, when selecting $\tau_c$ and $\tau_f$, consider the trade-off between compression rate and post-quantization model performance to meet different accuracy requirements across various scenarios.

- In addition, we plan to use the proposed coarse-to-fine proxy to determine whether specific features at a finer granularity are better suited for SQ or VQ quantization, such as channel-level or block-level finer granularity.

---

**Algorithm 1** RWKV Model Quantization Algorithm

---

**input** RWKV model with weights $\theta$, number of weights $M$, hyperparameters $C$ and $F$
**output** Quantized model with weights $\theta'$
1: Initialize: $C\_List \leftarrow [], F\_List \leftarrow [], \phi \leftarrow \{\}$ {Compute $\tau_c$}
2: **for** $m = 0 \ M - 1$ **do**
3:     Apply Equations (5) and (9) to $\theta_m$ to get $P_c$
4:     $C\_List.append(P_c)$
5: **end for**
6: $\tau_c \leftarrow \text{topK}(C\_List, k = C \cdot M, \text{largest} = \text{False})$
    {Compute $\tau_f$}
7: **for** $m = 0 \ M - 1$ **do**
8:     **if** $C\_List[m] < \tau_c$ **then**
9:         Apply Equations (10) and (17) to $\theta_m$ to get $P_f$
10:         $F\_List.append(P_f)$
11:     **end if**
12: **end for**
13: $\tau_f \leftarrow \text{topK}(F\_List, k = F \cdot C \cdot M, \text{largest} = \text{False})$
    {Perform quantization}
14: **for** $m = 0 \ M - 1$ **do**
15:     **if** $P_c \geq \tau_c$ **then**
16:         Apply Vector Quantization (VQ)
17:     **else**
18:         **if** $P_f \geq \tau_f$ **then**
19:             Apply Vector Quantization (VQ)
20:         **else**
21:             Apply Scalar Quantization (SQ)
22:         **end if**
23:     **end if**
24: **end for**
25: **return** $\theta' = 0$

---

