# OpenReview forum: "RWKVQuant: Quantizing the RWKV Family with Proxy Guided Hybrid of Scalar and Vector Quantization"
_ICML.cc/2025/Conference — ICML 2025 poster_

### Official Review · Reviewer_RAuy · 2025-02-23

**Overall Recommendation:** 3

**Summary:**

This paper proposed RWKVQuant, a method for PTQ of RWKV. They propose a hybrid method combining both scalar and vector quantization, and a decision rule for assigning different layers to these two methods. For appropriate values of hyperparameters (such as $\tau_c$ and $\tau_f$), they demonstrate strong performance on a wide range of benchmarks.

**Claims And Evidence:**

The claims are supported by clear evidence, but the setting of hyperparameters needs to be addressed more carefully.

**Essential References Not Discussed:**

I am not aware of essential references not discussed.

**Experimental Designs Or Analyses:**

I did not check the experimental designs.

**Methods And Evaluation Criteria:**

The benchmark datasets make sense for the problem at hand. However, much more sensitivity analyses are needed for the setting of hyperparameters.

**Other Comments Or Suggestions:**

* The short title has not been changed from "Submission and Formatting Instructions for ICML 2025"
* typo: line 097: "hight-order"
* typo: line 066 (second column): "we propose RWKVQuant, which **enhance** VQ..."
* typo: line 088 (second column): "the current word $x_t$ **and** can be derived by..."
* typo: line 202: "dose" should be "does"

**Other Strengths And Weaknesses:**

**I am very concerned about the setting of hyperparameters in this paper**.

The hyperparameters I'm most concerned about are $\tau_c$ and $\tau_f$ , as well as $K$ from equation 17. I would like to see a detailed section in the Appendix discussing in detail all decisions about how to set these hyperparameters in the different experiments, and detailed ablation studies / sensitivity analyses showing what happens to the performance of RWKVQuant as different values for these hyperparameters ($\tau_c$, $\tau_f$, and $K$ are used).

Appendix A.5 is an OK start, but it should have been explicitly pointed to in the main section. There needs to be a sentence somewhere in Section 3 saying that "the performance of RWKVQuant can vary depending on the setting of $\tau_f$ and $\tau_c$, see Appendix A.5".

I am also concerned by the admission of the following:
> Therefore, in future work, we will further explore how to determine appropriate values for $\tau_c$ and $\tau_f$.

In my opinion, the derivations in Section 3 (Method) are not sufficiently principled to warrant publication on their own, and so the merit of this work is based on the strong practical and empirical performance of RWKVQuant in Section 4 (Experiments). However, **a practical method is only useful if its hyperparameters can be set robustly.** I cannot recommend publication of this work until the bullet points laid out by the authors at the end of Appendix A.5 are satisfactorily addressed. I believe the answers are vital to widespread use of this method.

**Questions For Authors:**

1. > For both vision and language tasks, we select 128 samples from the corresponding test datasets for calibration

1a. Do you think that choosing the samples from the **test** dataset corresponds to a slight form of test set leakage? I would like to see an ablation where the samples are chosen from a different dataset (or at least where those 128 samples that are used for calibration are not used to monitor performance)

1b. To clarify: does "calibration" here mean the procedure for setting $\tau_c$ and $\tau_f$ such that 90% of the layers use SQ and 10% of the layers use VQ? I would like to see an ablation where this percentage of layers using SQ vs VQ is changed and the performance of RWKVQuant is tracked).

2. Lines 316-7 (second column) say
> For example, in RWKV7, $\tau_c$ is set to 1.54, while $\tau_f$ is set to 30.

However, Figure 5 is constructed
> Under the settings of $\tau_c=1.5$ and $\tau_f=50$.

Why is there this discrepancy?

**Relation To Broader Scientific Literature:**

PTQ of recurrent models has been much more difficult than that of Transformers, likely because recurrent computation allows errors to compound. It is very impressive that RWKVQuant is able to attain such strong performance in light of the known difficulty of this problem in the literature.

**Theoretical Claims:**

Section 3 is not so much a theoretical claim, but rather an explanation for the development of a heuristic that the authors then use.

---

> ### Author Rebuttal · Authors · 2025-04-01
>
> We sincerely thank you for your valuable time and efforts in reviewing our manuscript. We have addressed each comment and made the necessary revisions to improve the quality and clarity of our manuscript.
>
> >Concerned about the setting of hyperparameters.
>
> 1. For hyperparameters  $\tau_c$ and $\tau_f$
>
>     We have explicitly added the sentence "the performance of RWKVQuant can vary depending on the setting of $\tau_c$ and $\tau_f$, see Appendix A.5" to our latest manuscript after line 274 (first column).
>
> Actually, $\tau_c$ and $\tau_f$ are automatically set for each individual model and do not require adaptation. Specifically, we obtain their values in the following steps:
> - Compute the coarse-grained proxy $P_c$ (Eq.15) for each layer to be quantized.
> - Set $\tau_c$ to the value at the 50th percentile of all $P_c$.
> - Compute the fine-grained proxy $P_f$(Eq.17) for each layer whose $P_c$ < $\tau_c$.
> - Set $\tau_f$ to the value at the 20th percentile of all $P_f$.
>
> Although fine-tuning the percentile values for each network may further improve accuracy, the percentile values (i.e., 20% and 50%) —used in all our experiments—already delivers strong performance across all RWKV networks, as shown in Table 2 of the manuscript.
> We have also included more results in various configurations in our latest manuscript. Here is an example of RWKV-6-1B  in [Table R1](https://anonymous.4open.science/r/RWKVQuant-E375/Table_for_ablation_study.MD).
>
> 2. K represents the order of the Taylor expansion, we set it to 4 for all models.
>
> A larger number of K offers a closer result to the actual value (i.e., K=+∞). Experimentally, we find that K=4 is an ideal trade-off between accuracy and quantization cost under various $\tau_c$ and $\tau_f$. For instance, in RWKV-6-1B [Table R2](https://anonymous.4open.science/r/RWKVQuant-E375/Table_for_ablation_study.MD).
>
> >Writing Errors.
>
> Thanks for pointing them out. We have fixed them in our latest manuscript.
>
> >Concerns about calibrating from the test dataset.
>
> 1. RWKVQuant does not include any training process. All samples that are used for calibration are not used to monitor performance. They are only used to generate activations, rather than supervising.
>
> 2. For a fair comparison, we randomly sample 128 sequences from Lambda, following previous work like GPTQ [1] and GPTVQ [2]. In addition, we evaluate on various datasets that are not used for calibration, as shown in Table 2 in our manuscript.
> To further address your concern, we provide [Table R3](https://anonymous.4open.science/r/RWKVQuant-E375/Table_for_ablation_study.MD), where calibration is performed using only a small subset (128 samples) of the training dataset under the same conditions. The results indicate that our approach still maintains an advantage.
>
> >Does "calibration" here mean the procedure for setting $\tau_c$ and $\tau_f$ such that 90% of the layers use SQ and 10% of the layers use VQ?
>
> Calibration and proportion are two distinct concepts. Calibration is an common practice of PTQ, which samples data and feed it to models to generate activations. Making use of such activations, PTQ can optimize the weight quantization [1,2]. As for the proportion of SQ and VQ, it is fixed to our default configuration in this work corresponding to the first reply.
>
> >I would like to see an ablation where this percentage of layers using SQ vs VQ is changed and the performance of RWKVQuant is tracked.
>
> Please refer to the [Table R1](https://anonymous.4open.science/r/RWKVQuant-E375/Table_for_ablation_study.MD).
>
> >Why is the setting of $\tau_c$ and $\tau_f$ are not the same?
>
> The values of $\tau_c$ and $\tau_f$ can vary across different models. This variation occurs because these values are dictated by the percentile of the proxy specific to each individual model.
>
> References:
>
> [1] GPTQ: Accurate Post-Training Quantization for Generative Pre-trained Transformers.
>
> [2] GPTVQ: The Blessing of Dimensionality for LLM Quantization.

---

> > ### Comment · Reviewer_RAuy · 2025-04-04
> >
> > Thank you for your additional results on setting hyperparameters. I am satisfied that the method is reasonably robust. Please include all tables from rebuttals in the final manuscript. I am glad that you are also including an explicit pointer to Appendix A.5. I am raising my score to a 3.

---

> > > ### Author Response · Authors · 2025-04-07
> > >
> > > Dear Reviewer,
> > >
> > > Thank you for your constructive comments and valuable suggestions. We will include all the tables from rebuttals in the final manuscript.
> > >
> > > We are glad to address all your questions and sincerely appreciate your recognition.
> > >
> > > Best, Authors

---

### Official Review · Reviewer_D8uK · 2025-03-10

**Overall Recommendation:** 3

**Summary:**

This paper aims at reducing the memory usage and inference latency of RWKV through post-training quantization (PTQ) techniques. However, authors find that  non-linear operators and  larger amount of uniformly distributed weights hinder the effectiveness of previous PTQ methods. Therefore, authors propose RWKVQuant to quantize weights of RWKV models to extremely low-bit with less than 1% accuracy loss,  2.83$\times$ memory saving and 2.14$\times$ acceleration rate, which includes a coarse-to-fine proxy to adaptively select weight outliers and a codebook optimization algorithm to further enhance the performance of quantized RWKV models.

**Claims And Evidence:**

Claims in the "introduction" section is supported by Figure 1&5 and Table 1.

**Essential References Not Discussed:**

All necessary references are discussed.

**Experimental Designs Or Analyses:**

I've checked all experimental settings, comparison and results in this paper. See "Other Strengths And Weaknesses" part of this review for my major & minor concerns about the experimental part.

**Methods And Evaluation Criteria:**

I've checked all theoretical and qualitative analysis and claims in this paper. See "Other Strengths And Weaknesses" part of this review for my major & minor concerns about the methodology and equation derivation.

**Other Comments Or Suggestions:**

See "Minor weaknesses" part above.

**Other Strengths And Weaknesses:**

## Strengths:
1. The bottleneck of quantizing RWKV and the differences between RWKV models and Transformer-based LLMs are both clearly claimed and explained. Thus the motivation of this paper is clear and well-explained.
2. Figures are fancy and easy to follow, all legends in figures are clear and captions provide detailed information about the corresponding figure.
3. Preliminary is concise. The process of proposed method is clearly described, although the symbol is a little bit confusing.
4. Experiments are extensive and prove the effectiveness of the methods proposed in this paper.


## Major weaknesses:
1. In section 2.2, authors introduce SQ and VQ, but fail to claim the pros and cons of these two techniques. Therefore, I'm a little bit confused by why authors choose to use hybrid quantization instead of only use VQ since the outliers in applying SQ to RWKV is unsolvable.
2. Eq. 11 is weird, since $$\\sum_{i=1}^n G'_i = 1$$

and $$\\sum_{i=1}^n \\frac{1}{n} = 1$$, which means Eq. 11 is always satisfied. Maybe authors mean

$$\\sum_{i=1}^n \\delta= \\sum_{i=1}^n (G'_i - \\frac{1}{n}) = 0$$, which means conducting the element-wise subtraction first then conducting the sum operator.

3. Why the $s_k$ in Eq. 15 can be omitted? if so, the Taylor expansion won't be satisfied, then the $P_f(G')$ does not equal to the k-th expansion of $P_c(G')$. Maybe authors can add some experimental comparison of w/ & w/o $s_k$ for double-check and further clarification since $s_k$ is easy to calculate.
4. Seems like the fine-grained proxy is just a Taylor expansion version of coarse-grained one. What if just set a smaller $\\tau_c$? As shown in Figure 3 (b) & (c), if $\\tau_c$ is set as 0.95 (0.94(Fig. 3c) <0.95 <0.96(Fig. 3b)) and w/o $\\tau_f$, then it will also utilizes VQ when the condition depicted in (b) happened. Authors should add more comparison about different setting of $\\tau_c$ and w/ w/o $\\tau_f$.
5. The section of "codebook optimization for element-wise multiplication" seems like just applying percentile-based clipping operation into a traditional VQ, instead of simply average all samples, thus solve the outlier phenomenon. This part lacks novelty. It would be better to further explain the novelty and differences between authors' method and previous arts.
6. (1). Why the setting of bpw of previous methods are 3.25/3.5, while authors' method is 3.275? Just for ensuring that SQ is used for 9/10 of the layers and VQ for 1/10? How the performance changed with different settings of proportion of SQ and VQ? Lack relevant experiments and analysis. (2). Meanwhile, results in Table 2&3 both prove that the performance of VQ method GPTVQ is much better than all other previous work, is it because VQ methods often provide better performance with latency overheads? Then authors should also compare the flops and performance with different settings of proportion of SQ and VQ.
7. Authors only provide results with bpw around 3-bit, how about other bit-width, e.g. 2-bit & 4-bit.
8. As shown in Table 11, it seems like the effectiveness of codebook optimization is much weaker with larger-scale RWKV on Wiki2. I'm curious about this phenomenon, is it because with the model scale becomes larger, the outliers affect the performance less significantly? Maybe  authors can explain this part shortly.

## Minor weaknesses:
1. Symbol conflict in Eq. 6: since subscript $i$ is used in the numerator $G_i$, the subscript in denominator $\sum_{i=1}^n G_i$ should be a new one, which can be formulated as $\sum_{j=1}^n G_j$.
2. The explanation of superscript $k$ in Eq. 12 is missing. If I'm not missing anything, does it represent k-th  order partial?
3. Symbol reuse: $m$ in section 3.1 represents index of number of weights, while in section 3.2 & 2.2 represents shape of a tensor or weight $\\mu$. .

**Questions For Authors:**

See above.

**Relation To Broader Scientific Literature:**

All contributions are technical and all datasets used for experiments are open-sourced. Thus no key contributions of this paper related to the broader scientific literature.

**Theoretical Claims:**

I've checked all theoretical and qualitative analysis and claims in this paper. See "Other Strengths And Weaknesses" part of this review for my major & minor concerns about the methodology and equation derivation.

---

> ### Author Rebuttal · Authors · 2025-04-01
>
> We sincerely thank you for your valuable time and efforts in reviewing our manuscript. We have addressed each comment and made the necessary revisions to improve the quality and clarity of our manuscript.
>
> >Why authors choose to use hybrid quantization instead of only use VQ since the outliers in applying SQ to RWKV is unsolvable.
>
> VQ and SQ are suitable for differently distributed weights. VQ excels in managing outliers but underperforms when dealing with uniform distributions. For weights without outliers, SQ can achieve better results. Thus, we are motivated to hybrid them.
>
> >Concerns of Eq. 11.
>
> Thanks for pointing it out. We have fixed this problem in our latest manuscript.
>
> >Why the $s_k$ in Eq.15 can be omitted.
>
> Eq.15 is an equal expression of the Taylor expansion, but Eq.17 is obtained by performing an absolute operation to Eq.15. Thereby $s_k$ can be omitted. We explain the reason for this operation in step5 of the fine-grained proxy (line 241-245).
>
> >Concerns on the fine-grained proxy.
>
> Fine-grained proxy is not just a Taylor expansion version of coarse-grained one. It extracts the outliers features by performing an absolute function to specific terms of the Taylor expansion, as stated in step5 of the fine-grained proxy (line 241-245).
>
> >Authors should add more comparison about different setting of $\tau_c$ and w/ w/o $\tau_f$.
>
> Actually, $\tau_c$ and $\tau_f$ are automatically set for each individual model and do not require adaptation. Specifically, we obtain their values in the following steps:
> - Compute the coarse-grained proxy $P_c$ (Eq.15) for each layer to be quantized.
> - Set $\tau_c$ to the value at the 50th percentile of all $P_c$.
> - Compute the fine-grained proxy $P_f$ (Eq.17) for each layer whose $P_c$ < $\tau_c$.
> - Set $\tau_f$ to the value at the 20th percentile of all $P_f$.
>
> Although fine-tuning the percentile values for each network may further improve accuracy, the percentile values (i.e., 20% and 50%) —used in all our experiments—already delivers strong performance across all RWKV networks, as shown in Table 2 of the manuscript.
> We have also included more results in various configurations in our latest manuscript. Here is an example of RWKV-6-1B  in [Table R1](https://anonymous.4open.science/r/RWKVQuant-E375/Table_for_Tr_Tf_wo_f.MD).
>
> >It would be better to further explain the novelty and differences between the codebook optimization and previous arts.
>
> Firstly, our approach is designed for element-wise multiplication operations (widely-applied in RWKV), instead of matrix multiplication (widely-applied in Transformer-based LLMs). Compared to matrix level, element-wise poses a challenge of batch integration for calibration.
>
> Secondly, we discover that the activations of RWKV follow the Guassian distribution as shown in Figure 4. Thus, our codebook optimization can be well applied.
>
> Thirdly, directly applying previous arts to RWKV leads to significant accuracy decline as shown in Table 7 of the manuscript. Our method can effectively enhance performance.
>
> >How the performance changed with different settings of proportion of SQ and VQ?
>
> 1. VQ and SQ are suitable for differently distributed weights. VQ excels in managing outliers but underperforms when dealing with uniform distributions. For weights without outliers, SQ can achieve better results. Thus, we are motivated to hybrid them.
>
> 2. It is possible to improve accuracy by adjusting the percentiles for each network. However, this default configuration can already achieve effective results across all RWKV networks and typical tasks as shown in Table 2 of the manuscript.
>
> >Why are VQ methods much better than all other previous work?
>
> Generally, SQ is not good at handling outliers for its scaling characterstic, especially in aggressive bitwidth (e.g., lower than 4 bit). In contrast, VQ has the ability to capture the distribution of the original sequence and offers an advantage in terms of compression ratio [1].
>
> >Results on other bit-width settings.
>
> Please refer to the  Table R1 in the above response.
>
> >The effectiveness of codebook optimization is much weaker with larger-scale RWKV.
>
> Generally, larger-scale models are less sensitive to quantization, primarily due to their larger amount of redundant parameters as mention in [2,3]. Thus, the optimal quantization strategy may be less obvious for these models.
>
> >Symbol conflict in Eq. 6.
>
> Thanks for pointing it out. We have fixed this problem in our latest manuscript.
>
> >The explanation of superscript k in Eq. 12 is missing.
>
> We explain it next to Eq. 12 in line 188-190.
>
> >Symbol reuse: m.
>
> Thanks for pointing it out. We have fixed this problem in our latest manuscript.
>
> References:
>
> [1].  QuIP#: Even Better LLM Quantization with Hadamard Incoherence and Lattice Codebooks
>
> [2]. A Survey of Quantization Methods for Efficient Neural Network Inference
>
> [3]. LLM Inference Unveiled: Survey and Roofline Model Insights

---

### Official Review · Reviewer_iLYA · 2025-03-13

**Overall Recommendation:** 3

**Summary:**

RWKV is a modern RNN architecture that faces deployment challenges on resource-constrained devices. RWKVQuant, a post-training quantization (PTQ) framework, is proposed to address the limitations of applying existing quantization methods to RWKV models. RWKVQuant uses a coarse-to-fine proxy to adaptively select quantization approaches and optimizes codebook performance for element-wise multiplication. Specifically, RWKVQuant can quantize weights to approximately 3-bits with less than 1% accuracy loss, while providing up to a 2.14× speedup in inference

**Claims And Evidence:**

This paper presents strong claims about RWKVQuant’s effectiveness in improving post-training quantization for RWKV models, backed by experimental results. It demonstrates weight quantization to 3 bits with less than 1% accuracy loss and a 2.14× speedup, using a hybrid SQ-VQ approach guided by Information Entropy and higher-order moments. The method shows promising performance across language and vision tasks, with reduced memory and faster inference — supported by ablation studies. However, clearer comparisons with other leading quantization frameworks and more detailed performance metrics would further strengthen the claims.

**Essential References Not Discussed:**

None

**Experimental Designs Or Analyses:**

1. Choice of Datasets and Tasks: Uses LAMBADA (language) and ImageNet, Coco, ADE20K (vision), covering both NLP and CV tasks.

2. Quantization Methods and Baselines: Compares RWKVQuant against SQ methods (RTN, GPTQ) and VQ methods (K-Means, GPTVQ).  Comparisons are broad, but ensuring equivalent configurations (e.g., bit representation, group sizes) across methods is essential for fairness. More transparency on parameter choices (e.g., bpw = 3.25/3.5) would improve clarity.

3. Performance Metrics: Uses perplexity (PPL) for language and Top-1, Box AP, MIoU for vision tasks.
Discussion: Metrics are appropriate, but reporting should include mean performance across runs, standard deviation, or confidence intervals to account for random variations.

**Methods And Evaluation Criteria:**

### Proposed Methods:

1. RWKVQuant Framework: Combines Scalar Quantization (SQ) and Vector Quantization (VQ), leveraging RWKV’s architecture for better performance.
2. Coarse-to-Fine Proxy: Uses Information Entropy to handle weight uniformity and outliers, adapting the quantization approach to the weight characteristics.
3. Codebook Optimization: Tailors codebook generation to RWKV’s element-wise multiplication modules for improved efficiency.


### Evaluation Criteria:

1. Benchmark Datasets: Uses standard datasets like LAMBADA (language) and ImageNet, Coco, ADE20K (vision) for fair comparisons.
2. Performance Metrics: Measures perplexity (PPL) for language and Top-1 Accuracy, AP, and MIoU for vision tasks — suitable for assessing quantization impact.
3. Comparative Analysis: Benchmarks against existing SQ, VQ methods, and strong baselines to validate performance improvements.

**Other Comments Or Suggestions:**

Please refer to the weakness part.

**Other Strengths And Weaknesses:**

### Strengths:

1. Novel Hybrid Quantization Framework: RWKVQuant introduces a significant advancement by combining Scalar Quantization (SQ) and Vector Quantization (VQ) tailored specifically for RWKV models. This hybrid approach effectively addresses inefficiencies in previous methods, optimizing quantization for non-linear operators in RWKV.
2. Empirical Results: The experiments show impressive outcomes, such as less than 1% accuracy loss while quantizing RWKV-6-14B to about 3 bits and achieving a 2.14x speedup. These benchmarks validate the framework’s real-world applicability, making it highly relevant for deploying large models on resource-constrained devices.
Thorough Understanding of RWKV Architecture:
3. The paper demonstrates a deep understanding of RWKV’s unique characteristics, such as Time Mixing and Channel Mixing, and tailors the quantization strategy accordingly. This approach is more effective than generic quantization methods, ensuring optimal performance.
4. Clarity and Structure: The paper is well-organized, with a clear flow from the problem statement to the methodology, experiments, and discussions. The inclusion of figures like the accuracy-model size curve effectively communicates complex concepts, making it easy to follow for researchers and practitioners.
### Weaknesses:

1. Limited Comparison with State-of-the-Art: While RWKVQuant shows promising results, the paper would benefit from a broader comparative analysis with other advanced quantization methods, such as Adaptive Weight Quantization or mixed precision methods, to establish its relative strengths more clearly.
Concerns of Overfitting:

The favorable experimental results raise concerns about overfitting, particularly with the Lambada dataset. Acknowledging this limitation and conducting experiments on more diverse datasets or employing cross-validation would provide a clearer picture of the method's generalization capability.
Practical Implementation Guidance:

Although the methodology is clear, the paper lacks practical implementation details. Providing examples, pseudo-code, or insights into hyperparameter tuning and the application of the coarse-to-fine proxy would make the framework more accessible for practitioners.

**Questions For Authors:**

1. Can this method be applied to activation quantization as well?
2. In Table 4, why does the speedup increase with model size? With weight quantization alone, what measures should be taken to achieve actual speedup?
3. What is the actual cost of the codebook optimization step?
4. Regarding practical implementation, could you provide specific strategies for tuning the parameters τ_c and τ_f? Were any heuristic approaches or guidelines identified during your experiments that could assist other researchers in achieving optimal performance?

**Relation To Broader Scientific Literature:**

Post-Training Quantization (PTQ): Builds on prior PTQ methods, highlighting the poor performance of traditional SQ and VQ on RWKV, similar to challenges in other architectures.
Hybrid Quantization Strategies: Introduces a hybrid SQ-VQ approach, aligning with prior research suggesting hybrid methods improve quantization performance.
Model Efficiency: Focuses on efficiency for resource-constrained devices, achieving ~3-bit quantization with high accuracy, supporting findings on T-LLM deployment

**Theoretical Claims:**

This paper does not explicitly present formal proofs for theoretical claims; rather, it primarily focuses on empirical results and the introduction of the RWKVQuant framework. However, it does make several theoretical assertions and claims, particularly about the quantization methods and their effects on the RWKV models.

---

> ### Author Rebuttal · Authors · 2025-04-01
>
> We sincerely thank you for your valuable time and efforts in reviewing our manuscript. We have addressed each comment and made the necessary revisions to improve the quality and clarity of our manuscript.
>
> > The paper would benefit from a broader comparative analysis with other advanced quantization methods.
>
> 1. Directly applying SOTA methods on Transformer-based LLMs on RWKV models can lead to significant accuracy degration. For example, applying QuaRot [5] to LLaMA series has only 1% accuracy drop, while that for RWKV series is over 8%.
>
> 2. There are few works that apply quantization for RWKV models. We are the first to combine SQ and VQ, achieving effective performance on the RWKV family.
>
> >Concerns of Overfitting:
>
> The characteristic of PTQ is that it enables rapid quantization deployment with a small calibration dataset. Current SOTA methods of Transformer-based LLMs do not encounter this issue.
>
> We also care about the overfitting challenges on PTQ for RWKV.  We perform experiments on nine datasets, **which are not used for calibration (i.e., zero-shot)**. As shown in Table 2 of the manuscript, RWKVQuant does not overfit.
>
> >Lack of pseudo-code:
>
> Thanks for you adivce, we've added an [Algorithm](https://anonymous.4open.science/r/RWKVQuant-E375/Table_for_New_Method.MD) to our latest manuscript.
>
> >Can this method be applied to activation quantization as well?
>
> Considering that bandwidth is the main bottleneck of LLMs' inference speed, this work is put forward to tackle the challenges associated with weight quantization. Moreover, applying this quantization approach to activations represents a highly promising and valuable area of research. We intend to conduct in-depth explorations of this aspect in our forthcoming research endeavors.
>
> >In Table 4, why does the speedup increase with model size? With weight quantization alone, what measures should be taken to achieve actual speedup?
>
> The actual speed up can be effected by various factors, such as instruct latency, memory access, computational time, optimization of de-quantization kernal, and so on.
>
> Generally, memory access takes the most importance across all above factors, since the decode phase is typically memory-bound. With the model size increase, memory access will account for more latency. Considering that weight quantization can reduce this cost, the speed up should also increase with the model size.
>
> >What is the actual cost of the codebook optimization step?
>
> The codebook optimization is a lightweight **offline** process; for instance, it only takes about 15 minutes for a 70-parameter model.
>
> >Could you provide specific strategies for tuning the parameters $\tau_c$ and $\tau_f$?
>
> Actually, $\tau_c$ and $\tau_f$ are automatically set for each individual model and do not require adaptation. Specifically, we obtain their values in the following steps:
> - Compute the coarse-grained proxy $P_c$ (Eq.15) for each layer to be quantized.
> - Set $\tau_c$ to the value at the 50th percentile of all $P_c$.
> - Compute the fine-grained proxy $P_f$ (Eq.17) for each layer whose $P_c$ < $\tau_c$.
> - Set $\tau_f$ to the value at the 20th percentile of all $P_f$.
>
> Although fine-tuning the percentile values for each network may further improve accuracy, the percentile values (i.e., 20% and 50%) —used in all our experiments—already delivers strong performance across all RWKV networks, as shown in Table 2 of the manuscript.
> We have also included more results in various configurations in our latest manuscript. Here is an example of RWKV-6-1B  in [Table R1](https://anonymous.4open.science/r/RWKVQuant-E375/Table_for_New_Method.MD).
>
> References:
>
> [1] AWQ: Activation-aware Weight Quantization for LLM Compression and Acceleration.
>
> [2] GPTQ: Accurate Post-Training Quantization for Generative Pre-trained Transformers.
>
> [3] GPTVQ: The Blessing of Dimensionality for LLM Quantization.
>
> [4] VPTQ: Extreme Low-bit Vector Post-Training Quantization for Large Language Models.
>
> [5] QuaRot: Outlier-Free 4-Bit Inference in Rotated LLMs

---

### Official Review · Reviewer_qMhV · 2025-03-18

**Overall Recommendation:** 3

**Summary:**

This paper introduces RWKVQuant, a post-training quantization framework for the RWKV model family. The main contributions are: 1) revealing the limitations of existing Scalar Quantization (SQ) and Vector Quantization (VQ) methods on RWKV; 2) proposing a coarse-to-fine proxy strategy to guide the hybrid use of SQ and VQ; 3) optimizing the codebook for RWKV's unique element-wise multiplication operation. Extensive experiments show that the method can quantize RWKV-6-14B to about 3-bit with less than 1% accuracy loss while achieving 2.14x speedup.

## update after rebuttal
Thank you for the author's response. After referring to the author's response and the comments of other reviewers, I decided to keep my score.

**Claims And Evidence:**

The paper's main claims are well supported by both experimental and theoretical evidence:

1. Table 1 verifies the more uniform weight distribution characteristic of RWKV through clustering loss comparison
2. Figure 3 validates the effectiveness of the proxy strategy through visualization analysis
3. Tables 2-3 demonstrate the method's superiority through comprehensive comparative experiments

**Essential References Not Discussed:**

All major related works are discussed, with no significant omissions noted.

**Experimental Designs Or Analyses:**

The experimental design is comprehensive and well-structured:

1. Validation across multiple RWKV model scales
2. Comparison with existing mainstream quantization methods
3. Rich ablation studies
4. Both qualitative and quantitative analyses provided

**Methods And Evaluation Criteria:**

The methodology is sound and evaluation criteria are comprehensive:

1. Method design progressively develops solutions based on RWKV characteristics
2. Evaluation metrics include model performance indicators such as accuracy and perplexity
3. Considers practical metrics like memory usage and inference speed
4. Thorough validation on multiple benchmark datasets

**Other Comments Or Suggestions:**

None

**Other Strengths And Weaknesses:**

**Strengths:**

1. First systematic study of RWKV model quantization. First paragraph of Introduction explicitly states this is the first comprehensive quantization framework for RWKV family.
2. Novel and effective method design.
    - Table 2 shows superior performance across multiple models.
    - Ablation studies validate the necessity of each component
3. Rigorous theoretical analysis.
    - Complete mathematical derivation in Section 3
    - Intuitive visualization explanation in Figure 3
4. Comprehensive experimental validation.
    - Validation on 7 different scale models
    - Comparison of multiple evaluation metrics in Tables 2-3

**Weaknesses:**

1. Lack of theoretical guidance for threshold selection in proxy strategy. Only empirical setting of $\tau_c$ and $\tau_f$ mentioned in experimental section.
2. Insufficient analysis of optimal quantization strategies across different model scales. No related discussion in experimental section.

**Questions For Authors:**

Please refer to Strengths And Weaknesses.

**Relation To Broader Scientific Literature:**

Clear articulation of relationships with existing work.

**Theoretical Claims:**

The theoretical analysis is complete:

1. Detailed analysis of SQ and VQ limitations in RWKV
2. Complete mathematical derivation for the coarse-to-fine proxy strategy
3. Codebook optimization method designed based on element-wise multiplication characteristics

---

> ### Author Rebuttal · Authors · 2025-04-01
>
> We sincerely thank you for your valuable time and efforts in reviewing our manuscript. We have addressed each comment and made the necessary revisions to improve the quality and clarity of our manuscript.
> >Only empirical setting of $\tau_c$ and $\tau_f$ mentioned in experimental section.
>
> Sorry for causing this confusion. Actually, $\tau_c$ and $\tau_f$ are automatically set for each individual model and do not require adaptation. Specifically, we obtain their values in the following steps:
> - Compute the coarse-grained proxy $P_c$ (Eq.15) for each layer to be quantized.
> - Set $\tau_c$ to the value at the 50th percentile of all $P_c$.
> - Compute the fine-grained proxy $P_f$ (Eq.17) for each layer whose $P_c$ < $\tau_c$.
> - Set $\tau_f$ to the value at the 20th percentile of all $P_f$.
>
> Although fine-tuning the percentile values for each network may further improve accuracy, the percentile values (i.e., 20% and 50%) —used in all our experiments—already delivers strong performance across all RWKV networks, as shown in Table 2 of the manuscript.
> We have also included more results in various configurations in our latest manuscript. Here is an example of RWKV-6-1B  in [Table R1](https://anonymous.4open.science/r/RWKVQuant-E375/Table_for_Tr_Tf.MD).
>
>
> >Insufficient analysis of optimal quantization strategies across different model scales. No related discussion in experimental section.
>
> In line 347-353 of the manuscript, we have disscussed this point. Here is a takeaway for your convenience.
> As shown in Table 2 in original paper, on small-scale models, our method reduces the error by 14.5% compared to the SOTA; on larger-scale models, our method performs almost consistently with floating-point precision and reduces the error by 62.34% compared to the SOTA. The following table compares the average zero-shot accuracy between origin models and our quantized models. It can be observed that our method is only of slight accuracy decline across different model sizes .
>
> |Model |RWKV-6-1B|RWKV-6-3B|RWKV-6-7B|RWKV-6-14B|
> |:----:|:----:|:----:|:----:|:----:|
> | FloatingPoint|54.39%|58.32%|61.69%|63.65%|
> | FloatingPoint|51.69%|55.79%|60.19%|62.69%|

---

### Decision · Program_Chairs · 2025-05-01

**Decision:**

Accept (poster)

**Comment:**

This paper proposes a post-training quantization (PTQ) method for compressing pre-trained RWKV models. The authors first point out two limitations of popular PTQ methods for LLMs when applying them to RWKV models. Scaler quantization (SQ) methods relying on smoothing factors/rotation matrices lead to extra computational overhead because of non-linear operators of RWKV models, vector quantization (VQ) methods relying on codebook and vector clustering cannot well handle uniformly distributed weights occupying a large portion of weights in an RWKV model. Therefore, the authors argue that applying SQ or VQ individually is not optimal for RWKV model, and thus propose a hybrid SQ-VQ method called RWKVQuant. It leverages a coarse-grained proxy to evaluate weight uniformity and a fine-grained proxy to identify outliers, so that SQ and VQ can be combined to achieve better performance on RWKV models. Experiments on both vision tasks and language tasks are conducted to showcase the effectiveness of the proposed method.

The paper initially|finally got scores (3,3,3,2)|(3,3,3,3) by four reviewers, who mostly recognized the motivation, the basic idea and its performance. Meanwhile, the reviewers also raised some concerns about 1) method formulation and writing (a lot of errors and typos); 2) lack comparisons with some state-of-the-art methods; 3) lack experiments on more diverse bit-width settings, like 2/4-bit and mix-precision; 4) missing ablations on the choice of hyper-parameters, calibration samples, etc.; 5) some experiments on language tasks may have overfitting issue; 6) lack sufficient experimental details.

The authors provided detailed responses to these concerns. All reviewers acknowledged the authors' rebuttal. Reviewer qMhV recognized the rebuttal and kept the same score for weak accept, and reviewer RAuy also recognized the rebuttal and increased the score to weak accept, and the other two originally positive reviewers kept the same score for weak accept. The AC read the paper, the reviews, the rebuttal and the reviewers' feedback, and mostly agree with reviewers' assessment. As the novelty of the propose method is not very strong and the writing and ablations of paper need a lot of improvements, I recommend "weak accept" to this paper. The authors are encouraged to carefully consider the reviewers' comments/suggestions and their rebuttal in the final paper revision.